# ChainGPT: Dual-Reasoning Model with Recurrent Depth and Multi-Rank State Updates

**Yunao Zheng**  **Xiaojie Wang**[*]
Beijing University of Posts and Telecommunications (BUPT), Beijing, China

**Lei Ren**  **Wei Chen**
Li Auto Inc., Beijing, China

## Abstract

Large language models, constrained by the fixed-depth Transformer architecture, struggle to solve complex reasoning tasks in an end-to-end manner. Existing approaches, such as Chain of Thought, improve reasoning depth to some extent but rely heavily on natural language generation, with computational costs increasing rapidly as the length of the generated sequence grows. To address these limitations, we propose ChainGPT, a dual-reasoning model that shifts reasoning into latent computational space. Within each layer, ChainGPT employs multi-substep state updates combined with state-guided sparse attention, enabling deep local computation and efficient long-range modeling without quadratic costs. Across layers, recurrent depth approach iteratively refine latent states, supported by adaptive training and stopping strategies that balance reasoning depth against computational budget. Theoretically, we show that ChainGPT can, in principle, simulate general computation, and empirically it delivers consistent improvements over comparable models, including on reasoning tasks that remain challenging for existing systems. By unifying efficiency and reasoning ability, ChainGPT provides a principled foundation for next-generation language models.

## 1 Introduction

Large Language Models (LLMs) have demonstrated strong capabilities in natural language interaction, but their architectures remain fundamentally constrained in expressive power. From the perspective of computational complexity theory, the fixed depth of standard Transformers (Vaswani et al., 2017) places them within circuit classes such as $AC^0$ or $TC^0$ (Arora & Barak, 2009; Hahn, 2020), which limits their ability to reliably perform reasoning tasks that scale with problem size, including multi-step planning, symbolic manipulation, and combinatorial problem-solving requiring deep recursion. In other words, under finite precision, LLMs are not Turing complete and thus cannot solve tasks involving deep planning in a purely end-to-end manner.

To address these limitations, Chain of Thought (CoT) (Wei et al., 2022) has emerged as a mainstream approach, guiding models toward solutions through intermediate natural language steps. However, since reasoning is represented as discrete token sequences, it is inherently restricted by the expressiveness of language, making it difficult to capture nonlinear or parallel reasoning modes (Yao et al., 2023). Furthermore, as task complexity increases, the length of CoT outputs grows rapidly, introducing significant computational overhead.

Researchers have explored two major directions to address these challenges. The first focuses on improving reasoning strategies, such as Tree of Thoughts (ToT) (Yao et al., 2023), which employs tree search to explore more complex reasoning paths. The second focuses on improving underlying architectural efficiency. For example, RWKV-7 (Peng et al., 2025) leverages faster Recurrent Neural Network (RNN) designs to reduce computational cost, while Jamba (Lieber et al., 2024) and Samba (Ren et al., 2024) adopt hybrid architectures to balance reasoning quality with efficiency.

---

[*]Corresponding author: xjwang@bupt.edu.cn

Despite these advances, fundamental limitations remain. On the one hand, improved reasoning strategies can capture richer semantic structures but still rely on natural language, and their computational cost grows exponentially with the number of branches or samples. On the other hand, RNNs are more efficient but their finite-dimensional state matrices cannot faithfully retain detailed context, which hinders performance on tasks requiring precise memory (Jelassi et al., 2024). Current hybrid architectures are mostly sequential combinations that lack effective coordination: preserving global attention faces scalability bottlenecks, while restricting attention to local windows fails to resolve the problem of limited state matrix capacity.

These observations indicate that an effective solution must address two fundamental challenges at once: enabling deep reasoning beyond simple token generation, and sustaining computational efficiency while preserving long-range dependencies. To meet these challenges, we introduce ChainGPT, a novel architecture that shifts reasoning from explicit token-level processes into a latent computational space. By integrating multi-substep computation within layers and iterative refinement across layers, ChainGPT achieves both reasoning depth and efficiency, maintaining near-linear complexity.

Intra-layer reasoning is realized through the Chain-Block, which integrates a state layer and an attention layer. In the state layer, we employ RWKV-Product, a LoRA-based multi-substep update mechanism that enables the model to perform multiple state updates within a single forward pass. Since each substep is low-rank, the computation remains highly efficient, while their cumulative effect can approximate high-rank transformations, allowing this layer to significantly enhance the expressiveness of state updates without incurring substantial computational overhead. To ensure the model can retain comprehensive contextual information, the attention layer introduces State-Guided Sparse Attention (SGSA). Unlike other RNNs that compress the entire history into a finite state, SGSA only attends to recent tokens within sliding windows and a set of periodic anchors. As the sequence grows, contextual information is progressively written into the state, enabling the state space to scale naturally with length and distributing memory across block-level fragments. Through this approach, the model maintains near-linear complexity while still guaranteeing effective recall of long-range information.

Inter-layer reasoning employs recurrent depth approach (Geiping et al., 2025), dividing the model into a bottom-level feature extractor, a recursive reasoning core, and a top-level output layer. The recursive core iteratively refines latent states until convergence, while training stability is ensured through a two-phase "gradient-free warmup + truncated backpropagation" strategy with entropy-based early stopping, offering controllable trade-offs between reasoning depth and compute.

Theoretically, ChainGPT is Turing complete and can model any computable function. Empirically, it solves problems that remain challenging for state-of-the-art LLMs, even without pretraining or CoT supervision. Across benchmarks, ChainGPT consistently outperforms comparably sized baselines on diverse tasks.

The main contributions of this work are summarized as follows:

1. We propose the ChainGPT architecture, introducing a dual-reasoning mechanism for language model reasoning. ChainGPT combines intra-layer deep computation with inter-layer iterative refinement, while maintaining near-linear complexity.

2. We introduce RWKV-Product, a LoRA-based multi-substep state update mechanism that significantly enhances intra-layer reasoning depth and the effective rank of state updates with minimal parameter overhead. And State-Guided Sparse Attention (SGSA), which builds long-range dependency paths with near-linear complexity and effectively resolves the state bottleneck problem of RNN.

3. By combining recurrent depth approach with intra-layer sub-step reasoning, ChainGPT shifts iterative reasoning from token generation to multi-round latent optimization, enabling dual recursive reasoning within the model. We prove theoretically that ChainGPT is Turing complete under ideal conditions, and experiments show that it surpasses existing methods on some challenging reasoning tasks and consistently outperforms baseline models of comparable scale on benchmark evaluations.

## 2 RELATED WORK

### 2.1 EXPRESSIVE BOTTLENECKS OF RNN MODELS

Although RNNs offer efficiency advantages, theoretical studies reveal a fundamental trade-off between expressiveness and computational feasibility. Merrill et al. (2024) showed that while linear RNNs with full-rank transition matrices can, in theory, recognize any regular language, their training costs are prohibitively high, limiting practical applicability. Grazzi et al. (2025) further demonstrated that under finite numerical precision, RNNs with diagonal transition matrices (e.g., Mamba (Gu & Dao, 2023)) fail to perform even basic state-tracking tasks. This observation was empirically validated by Jelassi et al. (2024), who found that RNNs exhibit significant performance bottlenecks in tasks such as multi-hop reasoning.

To balance efficiency and expressiveness, RWKV-7 introduced a "diagonal + rank-1" structural refinement, which achieved modest performance gains without substantial computational overhead. However, since each update is restricted to rank-1, its expressive power remains limited. DeltaProduct (Siems et al., 2025) attempted to address this issue through multi-step Householder updates, but the additional parameter overhead makes it impractical. The RWKV-Product proposed in this work introduces a LoRA-based multi-substep decoupled state update mechanism. Within a single forward pass, it executes multiple sub-steps, enabling "multiple rounds of reasoning within one layer." With only a small increase in parameters, the effective rank of updates can be dynamically extended to $M$ (a tunable hyperparameter), resulting in faster convergence and stronger representational power.

### 2.2 DIFFERENT DESIGN PRINCIPLES OF HYBRID ARCHITECTURES

Given the respective limitations of pure RNN and pure Transformer architectures, recent years have seen the emergence of various hybrid models that attempt to combine the strengths of both.

Jamba adopts a 1:7 alternating stack of Mamba and Transformer layers, retaining full self-attention mechanisms. While this approach excels in long-sequence tasks, the embedded $O(N^2)$ self-attention modules become the primary scalability bottleneck. For instance, at a 32k context length, the computation of a single attention layer already exceeds that of seven Mamba layers combined. In contrast, ChainGPT employs a sparse attention design, reducing complexity to $O(N(W + N/G))$, where $W$ denotes window size and $G$ the global anchor interval, thereby enabling near-linear scalability.

Samba alternates between Mamba and sliding-window attention (SWA) to achieve hybrid modeling. However, pure window-based mechanisms still face state bottlenecks. Our experiments confirm that for long-context tasks (>8k), pure SWA architectures exhibit degraded performance, whereas the global anchor mechanism introduced in ChainGPT completely resolves this issue.

### 2.3 REASONING METHODS BEYOND FIXED DEPTH

Research aimed at enhancing the reasoning ability of fixed-depth Transformers can be broadly divided into two categories: generation-based reasoning extensions and internal state-based reasoning enhancements.

The former is exemplified by CoT. CoT and its variants extend effective computational depth by generating structured reasoning steps. ToT generalizes this idea into tree search, enabling the exploration of multiple reasoning paths. Self-Consistency (Wang et al., 2022) improves reliability by ensembling multiple reasoning chains through voting. However, these methods face inherent limitations: computational cost is tightly coupled with generation length, and the reasoning process is constrained by the expressive power of natural language.

The latter category focuses on iterative computation within the model's latent state space. Universal Transformer (Dehghani et al., 2019) introduced depth-adaptive mechanisms that allow different positions to perform varying numbers of computational steps. Looped Transformer (Liu et al., 2023) placed the entire model in a loop, effectively increasing depth via multiple forward passes. More recent work explores recursion in Transformers, such as the Hierarchical Reasoning Model (HRM) (Wang et al., 2025), which performs iterative computation across hierarchical levels, and the Depth-Recurrent Model, which refines the Looped Transformer by proposing recurrent depth approach.

ChainGPT belongs to the category of internal-state reasoning but innovatively introduces a dual-reasoning mechanism: RWKV-Product performs multi-substep state updates within a single forward pass, explicitly enhancing intra-layer reasoning depth; recurrent depth approach, combined with truncated backpropagation and adaptive early stopping, achieve a stable trade-off between reasoning depth and computational budget.

# 3 CHAIN-BLOCK

## 3.1 OVERALL DESIGN

ChainGPT is composed of multiple stacked Chain-Blocks. Each block consists of two core sub-structures: a state layer and an attention layer, as illustrated in Figure 1.

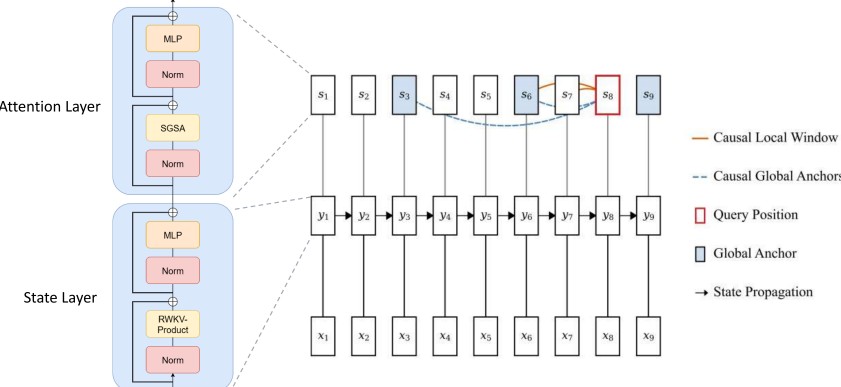

Figure 1: Chain-Block

**State Layer**: The core of the state layer is the proposed RWKV-Product mechanism. It employs a "multi-substep reasoning" approach, in which multiple substeps are executed within a single forward pass. Each substep maintains its own key-value representations and gating strategy. By adjusting the number of substeps, the model can explicitly increase its intra-layer reasoning depth without adding additional layers.

**Attention Layer**: Based on the output of the state layer, SGSA attends only to tokens within a local window and anchor tokens sampled at fixed intervals. By leveraging context summaries generated through multi-substep reasoning, SGSA constructs sparse yet globally covering dependency connections.

This design of "multi-step reasoning + sparse aggregation" enables ChainGPT to achieve deep reasoning within a single layer at near-linear computational cost, while efficiently modeling long-range semantic dependencies across layers.

## 3.2 RWKV-PRODUCT

Let $\mathbf{x}_t$ denote the input at step $t$, and $\mathbf{s}_t \in \mathbb{R}^{D \times D}$ the state matrix ($D$ is the input dimension):

$$\mathbf{s}_t = A(\mathbf{x}_t)\mathbf{s}_{t-1} + B(\mathbf{x}_t). \tag{1}$$

Here $A(\mathbf{x}_t) \in \mathbb{R}^{D \times D}$ is the transition matrix, and $B(\mathbf{x}_t)$ the injection matrix. Each sub-step $j$ has its own $A_j(\mathbf{x}_t)$ and $B_j(\mathbf{x}_t)$, so

$$A(\mathbf{x}_t) = \prod_{j=0}^{M-1} \left( \mathrm{diag}(\mathbf{a}_{t,j}) - \beta_{b,j}\, \mathbf{k}_{t,j}^{(b)} \mathbf{k}_{t,j}^{(b)\top} \right), \tag{2}$$

where $\mathrm{diag}(\mathbf{a}_{t,j})$ applies channel-wise decay, and $\beta_{b,j}$ controls the strength of the subsequent rank-1 correction. With independent update vectors, $A(\mathbf{x}_t)$ exhibits a "diagonal + rank-$M$" structure.

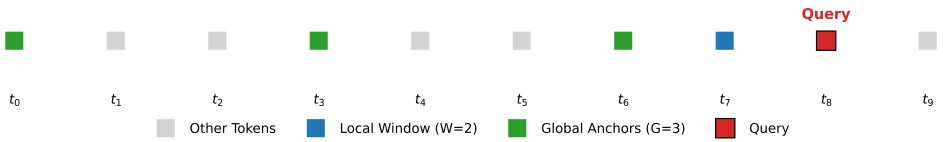

Figure 2: State-Guided Sparse Attention mechanism

The injection matrix aggregates sub-step contributions:

$$B(\mathbf{x}_t) = \sum_{j=0}^{M-1} \Big( \prod_{k=j+1}^{M-1} A_{t,k} \Big) \big( \beta_{c,j} \, \mathbf{k}_{t,j}^{(c)} \mathbf{v}_{t,j}^{\top} \big), \tag{3}$$

where $\mathbf{k}_{t,j}^{(c)}$ and $\mathbf{v}_{t,j}$ denote c-path key and value vectors, and $\beta_{c,j}$ determines the weight of the sub-step contribution.

To balance expressiveness and parameter efficiency, all keys/values adopt a shared baseline plus LoRA increments:

$$k_{t,j}^{(b)} = k_t^{\text{base}} + x_t W_j^{(b,k1)} W_j^{(b,k2)}, \tag{4}$$

$$k_{t,j}^{(b)} = \frac{k_{t,j}^{(b)}}{\|k_{t,j}^{(b)}\|_2 + \epsilon}, \tag{5}$$

$$k_{t,j}^{(c)} = k_t^{\text{base}} + x_t W_j^{(c,k1)} W_j^{(c,k2)}, \tag{6}$$

$$v_{t,j} = v_t^{\text{base}} + x_t W_j^{(c,v1)} W_j^{(c,v2)}. \tag{7}$$

Step sizes are specific to each sub-step, with shared bias $\alpha_{0,j}$ between the b- and c-paths. The symbol $\sigma$ denotes the sigmoid activation function:

$$\beta_{b,j} = \sigma\big(\alpha_{0,j} + (\mathbf{x}_t W_{1,j}^{(b)}) W_{2,j}^{(b)}\big), \tag{8}$$

$$\beta_{c,j} = \sigma\big(\alpha_{0,j} + (\mathbf{x}_t W_{1,j}^{(c)}) W_{2,j}^{(c)}\big). \tag{9}$$

This multi-substep design strictly enhances representational capacity with increasing $M$. Proofs and details appear in Appendix A; implementation in Appendix D.1.

### 3.3 STATE-GUIDED SPARSE ATTENTION (SGSA)

To capture long-range dependencies, we introduce SGSA, which operates on state outputs. For a query position, the accessible key set consists of:

- **Local Window:** $W$ neighbors around the query
- **Global Anchors:** positions sampled every stride $G$

Thus, the Chain-Block decomposes retrieval into "write and pointer-read": RWKV-Product aggregates local fragments and propagates them to anchors, while SGSA retrieves content via sparse addressing (Figure 2).

Appendix B proves that with proper hyperparameters, Chain-Block solves the Multi-Query Associative Recall (MQAR) task (Arora et al., 2024) for arbitrarily long sequences: for any $(k_i, v_i)$, if $q_j = k_i$, the model retrieves $v_i$.

Compared with dense $O(T^2)$ attention, our method requires only $O(T(W + T/G))$ sparse links, enabling efficient global modeling. Implementation is provided in Appendix D.2.

## 4 RECURRENT DEPTH APPROACH

### 4.1 OVERALL DESIGN

The recurrent depth approach decomposes the model into three modules: a bottom-level feature extractor, a recursive reasoning core, and a top-level output module. Each is constructed from stacked

Chain-Blocks. As shown in Figure 3, the bottom-level module generates stable contextual representations, the recursive core conducts multi-round reasoning over them, and the top-level module maps the final states into the target output distribution. Typically, the bottom-level and top-level modules each contain a single block. A full algorithmic description is provided in Appendix D.3.

This design enables ChainGPT to achieve computational universality with sufficient memory and time, allowing it to simulate any Turing machine and thus model arbitrary computable functions. A formal proof is provided in Appendix C.

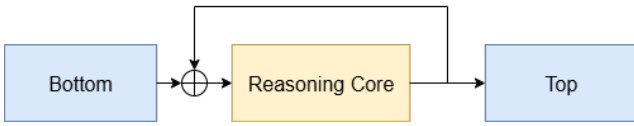

Figure 3: Recurrent depth approach

## 4.2 ADAPTIVE REASONING DEPTH

To reduce redundancy, an entropy-based early stopping mechanism monitors output stability. At iteration $t$, the core output is decoded:

$$\text{logits}_t = \text{Decode}(X_t), \quad \ell_t = \text{logits}_t[:, -1, :], \quad p_t = \text{softmax}(\ell_t). \tag{10}$$

**Stopping Criterion:** Entropy for sample $b$ is

$$H_t(b) = -\sum_{i=1}^{V} p_{t,i}(b) \log\big(p_{t,i}(b) + \varepsilon\big), \tag{11}$$

with difference

$$\Delta H_t(b) = H_{t-k}(b) - H_t(b). \tag{12}$$

If $\Delta H_t(b) \leq \tau$, recursion halts at step $t$. Threshold $\tau$ is a fixed constant.

## 5 EXPERIMENTS

This section validates the effectiveness of the ChainGPT architecture through comprehensive benchmarks, systematic ablations, and scalability analyses. Unless otherwise stated, all experiments are conducted on 8 NVIDIA L20 GPUs.

## 5.1 COMPREHENSIVE PERFORMANCE EVALUATION

We train ChainGPT models with 0.5B and 1.5B parameters, as well as baselines (Qwen2.5 (Qwen Team, 2024)), on 20B and 40B tokens from the FineWeb dataset (Penedo et al., 2024). Evaluation follows the `lm-eval-harness` framework (Biderman et al., 2024) under a zero-shot setting, covering six representative tasks on language understanding and commonsense reasoning (Clark et al., 2018; Zellers et al., 2019; Bisk et al., 2020; Welbl et al., 2017; Wang et al., 2018). As shown in Table 1, ChainGPT significantly outperforms baselines with the same parameter counts, with notable gains on ARC-Challenge and HellaSwag—tasks that require more complex reasoning—indicating stronger general-purpose reasoning ability.

Table 1: Comparison of comprehensive performance results

| Model | arc_challenge | arc_easy | hellaswag | piqa | sciq | glue | Avg. |
|---|---|---|---|---|---|---|---|
| Qwen2.5-0.5B | 0.2218 | 0.4082 | 0.3224 | 0.6425 | 0.5290 | 0.4664 | 0.4317 |
| ChainGPT-0.5B | **0.2389** | **0.4773** | **0.3644** | **0.6632** | **0.5330** | **0.4679** | **0.4575** |
| Qwen2.5-1.5B | 0.2696 | 0.5488 | 0.4091 | 0.6915 | 0.6380 | 0.4783 | 0.5059 |
| ChainGPT-1.5B | **0.2986** | **0.5779** | **0.4269** | **0.7018** | **0.6860** | **0.4836** | **0.5291** |

## 5.2 VALIDATION OF ARCHITECTURAL EFFECTIVENESS

**Effectiveness of the LoRA Sub-Step Mechanism**

Building on `BlinkDL/modded-nanogpt-rwkv` (BlinkDL, 2024), we extend RWKV-7 with a single LoRA sub-step to construct RWKV-Product. As shown in Table 2, with only a 0.1M increase in parameters, RWKV-Product reaches lower validation loss with fewer training steps, indicating that intra-layer multi-substep updates improve both training efficiency and representational capacity.

Table 2: Validation of the Effectiveness of the LoRA Sub-step Mechanism

| Model | Training Steps | Validation Loss |
|---|---|---|
| GPT2 | 19560 | $\approx 3.28$ |
| RWKV-7 | 3200 | 3.2715 |
| RWKV-Product | 2500 | 3.2684 |
| RWKV-Product | 3200 | 3.1901 |

**Impact of Sub-Step Count on Performance**

To study how the number of sub-steps affects performance, we evaluate models (180M parameters) trained with different sub-step counts $M$. Training uses 8B tokens from FineWeb and evaluation is performed on 1,000 validation samples. As shown in Figure 4, validation loss consistently decreases as $M$ increases, with $M=2$ offering a favorable trade-off. These results suggest that sub-steps effectively deepen intra-layer computation, thereby improving representational capacity.

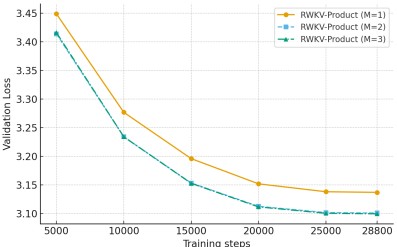

Figure 4: Effect of sub-step count on model performance

Figure 5: Comparison of coupled and decoupled state propagation paths

**Necessity of Decoupling State Propagation Paths**

We compare decoupled and coupled designs for state propagation. With approximately matched parameter counts, the decoupled variant consistently outperforms the coupled version throughout training, as shown in Figure 5.

**Hybrid Architecture Resolves Associative Recall Bottlenecks**

On the MQAR task, with head dimension = 64 and number of heads = 2, the pure State Space Model architecture (RWKV-7) exhibits noticeable accuracy degradation as sequence length grows, whereas ChainGPT with SGSA maintains accuracy above 99% across settings (Table 3). This indicates that SGSA's "anchor writing + sparse pointer retrieval" mechanism effectively mitigates the capacity bottleneck caused by finite state dimensionality.

Table 3: Performance on the MQAR task

| Model | (128,8) | (256,16) | (512,64) | (1024,128) | (2048,256) |
|---|---|---|---|---|---|
| RWKV-7 | > 99% | > 99% | 98.43% | 95.01% | 72.93% |
| ChainGPT | > 99% | > 99% | > 99% | > 99% | > 99% |

**Effectiveness of the Global Anchor Strategy**

We evaluate different global anchor intervals $G$ on PG-19 (Rae et al., 2019). As shown in Table 4, pure sliding-window attention (Samba approach) degrades beyond 8K context. In contrast, introducing only sparse periodic global anchors yields a perplexity trajectory that closely matches the much more expensive global attention (Jamba approach), with no notable degradation. This suggests that SGSA's "compression–aggregation" mechanism maintains long-range dependency modeling at near-linear complexity.

Table 4: Perplexity comparison for global anchor strategies

| Global Token Interval ($G$) | 2k | 4k | 6k | 8k | 10k | 12k | 14k | 16k |
|---|---|---|---|---|---|---|---|---|
| Global Attention | 17.78 | 16.82 | 16.46 | 16.20 | 16.11 | 16.07 | 16.07 | 16.08 |
| G=32 | 17.78 | 16.82 | 16.46 | 16.20 | 16.11 | 16.07 | 16.07 | 16.09 |
| G=64 | 17.78 | 16.82 | 16.46 | 16.20 | 16.11 | 16.07 | 16.07 | 16.08 |
| G=128 | 17.77 | 16.81 | 16.46 | 16.20 | 16.11 | 16.07 | 16.07 | 16.08 |
| Pure Window Attention | 17.78 | 16.83 | 16.47 | 16.21 | 16.12 | 16.09 | 16.10 | 16.12 |

**Impact of Recurrence Depth on Performance**

Keeping other conditions fixed, we increase the number of recursive iterations from $\times 1$ to $\times 16$. As shown in Figure 6, validation perplexity decreases steadily with more iterations and reaches its best at $\times 12$.

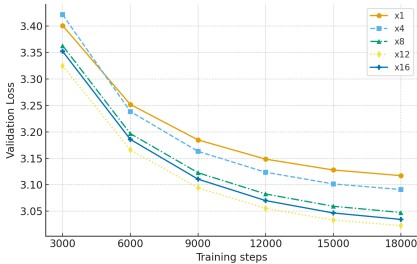

Figure 6: Effect of recurrence depth on model performance

**Comparison with Other Recurrent Models and CoT Methods**

To directly evaluate the performance of ChainGPT against other recurrent models and CoT methods, we conducted controlled arithmetic reasoning experiments on the public GOAT arithmetic dataset (Liu & Low, 2023) in Table 5. All models were trained from scratch under identical training–testing splits, tokenization schemes, and optimization settings to ensure fair comparison. The evaluated models include the baseline architectures Qwen3 and RWKV-7, their recurrent variants (Qwen3+Loop, RWKV-7+Loop), HRM, our proposed ChainGPT, as well as their CoT-augmented counterparts (Qwen3+CoT, ChainGPT+CoT).

Table 5: Accuracy on the GOAT dataset

| Model | Accuracy |
|---|---|
| Qwen3 | 33.82% |
| RWKV-7 | 23.69% |
| Qwen3 + Loop | 50.54% |
| RWKV-7 + Loop | 24.81% |
| HRM | 54.00% |
| **ChainGPT** | **57.53%** |
| Qwen3 + CoT | 88.43% |
| **ChainGPT + CoT** | **99.98%** |

Without CoT supervision, ChainGPT outperforms existing recurrent reasoning architectures, indicating that its intrinsic latent recursion mechanism enables more effective multi-step arithmetic

reasoning. When combined with explicit CoT guidance, ChainGPT+CoT significantly surpasses Qwen3+CoT, achieving nearly perfect accuracy. This result suggests that ChainGPT's implicit recursive states and the external CoT trajectory are complementary, jointly enhancing the model's reasoning capacity.

**Analysis on Challenging Reasoning Tasks**

To assess ChainGPT in complex reasoning scenarios, we conduct systematic evaluations on three representative hard tasks: ARC-AGI-1 (Chollet, 2019), Sudoku-Extreme, and Maze-Hard (Wang et al., 2025). Following HRM, we train 30M-parameter ChainGPT from scratch on 1000 samples, and train Direct Pred as a Transformer baseline under identical settings. Results are shown in Figure 7. For ARC-AGI-1, the scores are taken from the official leaderboard, while for Sudoku-Extreme and Maze-Hard, the scores are obtained through the corresponding APIs.

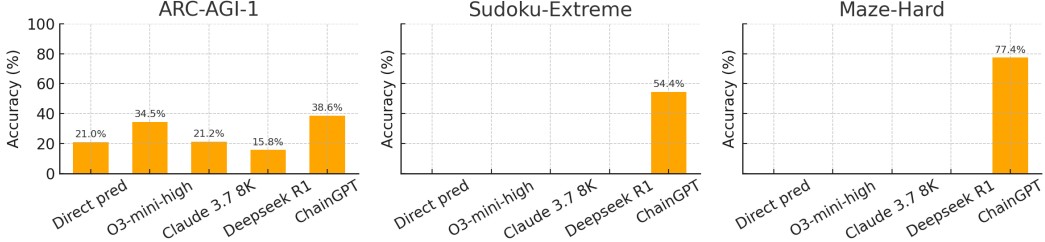

Figure 7: Benchmark accuracy comparison across different tasks

ARC-AGI-1 evaluates abstract rule induction under extreme few-shot conditions, where each problem provides only a handful of input–output examples. The central challenge lies in learning reusable compositional rules rather than memorizing patterns. Conventional deep learning and CoT-based approaches often fail to achieve stable generalization in such settings.

Sudoku-Extreme aggregates acknowledged hard Sudoku instances ranging from Kaggle datasets to Magictour-1465 and Forum-Hard/Extreme, with `tdoku` backtracking steps used as an objective difficulty measure. The difficulty stems from the need for precise constraint propagation and backtracking. Fixed-depth Transformers are not well-suited for long-trajectory backtracking, and parallel CoT sampling cannot reliably ensure deterministic correctness.

Maze-Hard requires finding shortest paths on $30 \times 30$ grids with enforced path lengths $> 110$. The challenge is to construct optimal solutions—mere reachability is insufficient. Large Transformers and CoT reasoning often generate suboptimal or invalid paths on large-scale mazes, showing their limitations in stable graph search.

ChainGPT excels in these scenarios due to its dual-reasoning mechanism, which enables multi-round, revisitable reasoning beyond conventional one-pass prediction or shallow CoT sampling. This unified design supports iterative rule induction and constraint propagation with contradiction checking, allowing the model to emulate algorithmic processes more faithfully across tasks. As shown in Figure 7, ChainGPT achieves 38.6% accuracy on ARC-AGI-1—surpassing strong baselines such as o3-mini-high and Claude-3.7-8K—54.4% accuracy on Sudoku-Extreme, and 77.4% accuracy on Maze-Hard with valid optimal solutions. These results suggest that ChainGPT effectively addresses the limitations of existing approaches and demonstrates robust performance on challenging reasoning tasks.

## 5.3 SPEED BENCHMARKING

We evaluate the computational efficiency of different modules on sequence lengths from 4K to 16K using a single NVIDIA A800 GPU, with total model dimension $= 1024$ and head dimension $= 64$, and attention implemented via F.scaled_dot_product_attention. Results in Figure 8 show that standard attention suffers from a quadratic scalability bottleneck as sequence length $N$ increases, whereas RWKV-Product achieves approximately linear runtime growth with controllable overhead across sub-step configurations, enabling an efficiency–performance trade-off. Furthermore, the hybrid architecture, which integrates state layers with sparse attention, demonstrates near-linear scaling

through a "state compression—sparse aggregation" strategy, offering clear efficiency advantages for long-range modeling under constrained computational cost.

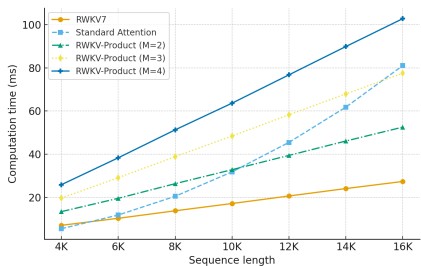

Figure 8: Computation efficiency comparison (ms)

## 6 CONCLUSION

In summary, ChainGPT introduces a principled dual-reasoning architecture that directly addresses the expressiveness bottlenecks of existing LLMs by shifting reasoning from token-level generation into latent computational space. Through the combination of RWKV-Product's multi-substep state updates and SGSA's near-linear global modeling, ChainGPT achieves intra-layer depth and efficient long-range reasoning simultaneously, while recurrent depth approach further extend its computational universality. Theoretical analysis establishes Turing completeness under proper configurations, and extensive experiments confirm that ChainGPT not only surpasses state-of-the-art baselines of comparable scale but also solves reasoning tasks that remain unsolved by current LLMs. These results highlight ChainGPT's unique advantage in unifying efficiency, scalability, and reasoning power, opening new directions for the design of next-generation language models.

## 7 ACKNOWLEDGMENTS

The work is supported by Beijing Natural Science Foundation (L247010) and National Key RD Program of China (No. 2024YFF0907003). We are grateful to the anonymous reviewers for their valuable and encouraging feedback and suggestions on this study.

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

## A  THEORETICAL ADVANTAGES OF RWKV-PRODUCT'S EXPRESSIVE POWER

This appendix presents a rigorous theoretical analysis of the multi-substep update mechanism used in RWKV-Product. Our goal is to show, mathematically, that as the number of sub-steps $M$ increases, both the family of realizable linear transformations and the capacity of information injection strictly expand. This analysis provides a solid mathematical foundation for the advantages of multi-substep updates over traditional single-step updates and supports the design's rationale.

To keep notation consistent, we first define the update operator for each sub-step. Let $D_{t,j} := \mathrm{diag}(\mathbf{a}_{t,j}) \in \mathbb{R}^{D \times D}$ be a diagonal matrix. For simplicity, we absorb the step-size factors into the vectors by defining $\mathbf{u}_{t,j} := \sqrt{\beta_{b,j}}\mathbf{k}_{t,j}^{(b)}$ and $\mathbf{v}_{t,j} := \sqrt{\beta_{b,j}}\mathbf{k}_{t,j}$. Hence, at time $t$, the transformation matrix of the $j$-th sub-step has the following "diagonal minus rank-1" form:

$$A_{t,j} = D_{t,j} - \mathbf{u}_{t,j}\mathbf{v}_{t,j}^\top \tag{13}$$

**Basic Assumption:** To ensure the validity of subsequent derivations, we assume all diagonal matrices $D_{t,j}$ are invertible; i.e., no diagonal entry is zero.

### A.1  LOW-RANK STRUCTURE OF PRODUCTS: COMPACT WY REPRESENTATION

**Theoretical significance:** Equivalently viewing the composite effect of $M$ successive rank-1 updates as a single "diagonal + rank-$M$" transformation $A(\mathbf{x}_t) = \widetilde{D} - \widetilde{U}\widetilde{T}\widetilde{V}^\top$ clarifies the algebraic structure underlying enhanced expressiveness: the upper bound on attainable rank increases from 1 to $M$.

**Lemma 1:** Given vector pairs $\{(\mathbf{u}_j, \mathbf{v}_j)\}_{j=0}^{M-1}$, let $\hat{A} = \prod_{j=M-1}^{0}(I - \mathbf{u}_j\mathbf{v}_j^\top)$. Then there exist matrices $U, V \in \mathbb{R}^{D \times M}$ and an upper-triangular matrix $T \in \mathbb{R}^{M \times M}$ such that

$$\hat{A} = I - UTV^\top, \quad \text{and} \quad \mathrm{rank}(UTV^\top) \leq M. \tag{14}$$

**Proof:** By induction. Let $\hat{A}_m = \prod_{j=m-1}^{0}(I - \mathbf{u}_j\mathbf{v}_j^\top)$. The base case $m = 1$ is immediate. Assume $\hat{A}_m = I - U_m T_m V_m^\top$ with $T_m \in \mathbb{R}^{m \times m}$ upper triangular. Consider $\hat{A}_{m+1} = \hat{A}_m(I - \mathbf{u}_m\mathbf{v}_m^\top)$:

$$\hat{A}_{m+1} = (I - U_m T_m V_m^\top)(I - \mathbf{u}_m\mathbf{v}_m^\top) \tag{15}$$

$$= I - U_m T_m V_m^\top - \mathbf{u}_m\mathbf{v}_m^\top + (U_m T_m V_m^\top \mathbf{u}_m)\mathbf{v}_m^\top \tag{16}$$

$$= I - [U_m, \mathbf{u}_m]\begin{bmatrix} T_m & -T_m(V_m^\top \mathbf{u}_m) \\ \mathbf{0} & 1 \end{bmatrix}\begin{bmatrix} V_m^\top \\ \mathbf{v}_m^\top \end{bmatrix}. \tag{17}$$

Define $U_{m+1} = [U_m, \mathbf{u}_m]$, $V_{m+1} = [V_m, \mathbf{v}_m]$, and the corresponding $(m+1) \times (m+1)$ upper-triangular $T_{m+1}$. The inductive step holds.

**Lemma 2:** Suppose each sub-step transformation is $A_j = D_j - \mathbf{u}_j\mathbf{v}_j^\top$. Then their product $A := \prod_{j=M-1}^{0} A_j$ can be written as

$$A = \widetilde{D} - \widetilde{U}\widetilde{T}\widetilde{V}^\top, \tag{18}$$

where $\widetilde{D} = \prod_{j=M-1}^{0} D_j$ and $\mathrm{rank}(\widetilde{U}\widetilde{T}\widetilde{V}^\top) \leq M$.

**Proof:** Again by induction. Let $P_m = \prod_{j=m-1}^{0} A_j$. The base case $m = 1$ is trivial. Assume $P_m = \widetilde{D}_m - L_m$, where $\widetilde{D}_m = \prod_{j=m-1}^{0} D_j$ and $\mathrm{rank}(L_m) \leq m$. Consider $P_{m+1} = A_m P_m = (D_m - \mathbf{u}_m\mathbf{v}_m^\top)(\widetilde{D}_m - L_m)$:

$$P_{m+1} = (D_m\widetilde{D}_m) - \underbrace{\left[D_m L_m + (\mathbf{u}_m\mathbf{v}_m^\top)\widetilde{D}_m - (\mathbf{u}_m\mathbf{v}_m^\top)L_m\right]}_{L_{m+1}}. \tag{19}$$

Let $\widetilde{D}_{m+1} = D_m\widetilde{D}_m$. The column space of the low-rank part $L_{m+1}$ is contained in $\mathrm{col}(D_m L_m) + \mathrm{span}(\mathbf{u}_m)$. Since $\mathrm{rank}(L_m) \leq m$, we have $\dim(\mathrm{col}(D_m L_m)) \leq m$, hence $\dim(\mathrm{col}(L_{m+1})) \leq m + 1$, i.e., $\mathrm{rank}(L_{m+1}) \leq m + 1$. The induction closes.

**Conclusion 1:** $A(\mathbf{x}_t) = \widetilde{D} - \widetilde{U}\widetilde{T}\widetilde{V}^\top$ with $\mathrm{rank}(\widetilde{U}\widetilde{T}\widetilde{V}^\top) \leq M$.

## A.2 STRICT INCLUSION: $\mathcal{F}_1 \subsetneq \mathcal{F}_M$ FOR $M \geq 2$

**Theoretical significance:** We prove, in a constructive manner, that multi-substep updates strictly enlarge the family of realizable linear transformations. Concretely, there exist matrices representable by a product of two "diagonal minus rank-1" factors ($M=2$), but not representable by any single-step mechanism ($M=1$). This establishes a set-theoretic notion of *strictly stronger* expressive power.

Define the matrix families (diagonal factors are assumed invertible, consistent with the basic assumption in Appendix A):

$$\mathcal{F}_M := \left\{ \prod_{j=M-1}^{0} \left(D_j - \mathbf{u}_j\mathbf{v}_j^\top\right) : \ D_j \text{ invertible diagonal} \right\}, \tag{20}$$

$$\mathcal{F}_1 := \left\{ D - \mathbf{u}\mathbf{v}^\top : \ D \text{ invertible diagonal} \right\}. \tag{21}$$

**Theorem 1:** For dimension $D \geq 3$ and sub-steps $M \geq 2$, we have the strict inclusion

$$\mathcal{F}_1 \subsetneq \mathcal{F}_M. \tag{22}$$

**Proof:** We show (i) $\mathcal{F}_1 \subseteq \mathcal{F}_M$, and (ii) there exists a matrix in $\mathcal{F}_2$ but not in $\mathcal{F}_1$.

*(i) Inclusion $\mathcal{F}_1 \subseteq \mathcal{F}_M$.* Take any $A \in \mathcal{F}_1$, so $A = D - \mathbf{u}\mathbf{v}^\top$ for some invertible diagonal $D$. Construct an element of $\mathcal{F}_M$ by setting

$$D_0 = D, \ \mathbf{u}_0 = \mathbf{u}, \ \mathbf{v}_0 = \mathbf{v}, \qquad D_j = I, \ \mathbf{u}_j = \mathbf{0} \ (j = 1, \ldots, M-1),$$

where $I$ is the identity matrix (invertible diagonal). Then

$$\prod_{j=M-1}^{0} \left(D_j - \mathbf{u}_j\mathbf{v}_j^\top\right) = (I)\cdots(I)\,(D - \mathbf{u}\mathbf{v}^\top) = A,$$

hence $\mathcal{F}_1 \subseteq \mathcal{F}_M$.

*(ii) Strictness via a constructive separation.* Proposition 1 below constructs an explicit matrix $A^\star \in \mathcal{F}_2$ such that $A^\star \notin \mathcal{F}_1$. Since $\mathcal{F}_2 \subseteq \mathcal{F}_M$ for all $M \geq 2$, this implies $\mathcal{F}_1 \subsetneq \mathcal{F}_M$.

**Proposition 1 (Constructive separation):** Let $D \geq 3$ and $\{e_i\}_{i=1}^D$ be the standard basis of $\mathbb{R}^D$. Define

$$A^\star := (I - e_2 e_3^\top)(I - e_1 e_2^\top) = I - e_1 e_2^\top - e_2 e_3^\top. \tag{23}$$

Then $A^\star \in \mathcal{F}_2$ but $A^\star \notin \mathcal{F}_1$.

**Proof:** The membership $A^\star \in \mathcal{F}_2$ is immediate: both factors are of the form

$$I - e_a e_b^\top \;=\; D - \mathbf{u}\mathbf{v}^\top \quad \text{with} \quad D = I, \; \mathbf{u} = e_a, \; \mathbf{v} = e_b,$$

where $D$ is invertible diagonal.

To show $A^\star \notin \mathcal{F}_1$, assume for contradiction that

$$A^\star = D - \mathbf{u}\mathbf{v}^\top$$

for some invertible diagonal $D$ and vectors $\mathbf{u}, \mathbf{v} \in \mathbb{R}^D$. For any $i \neq j$, the diagonal matrix contributes nothing to off-diagonal entries, hence

$$A_{ij}^\star = -(\mathbf{u}\mathbf{v}^\top)_{ij} = -u_i v_j, \qquad i \neq j. \tag{24}$$

From the explicit form $A^\star = I - e_1 e_2^\top - e_2 e_3^\top$, we have the off-diagonal pattern

$$A_{12}^\star = -1, \qquad A_{23}^\star = -1, \qquad A_{13}^\star = 0.$$

The equalities $A_{12}^\star = -u_1 v_2 = -1$ and $A_{23}^\star = -u_2 v_3 = -1$ imply

$$u_1 \neq 0, \; v_2 \neq 0, \quad \text{and} \quad u_2 \neq 0, \; v_3 \neq 0.$$

In particular, $u_1 \neq 0$ and $v_3 \neq 0$ together imply $u_1 v_3 \neq 0$, hence

$$A_{13}^\star = -u_1 v_3 \neq 0,$$

which contradicts $A_{13}^\star = 0$. Therefore $A^\star \notin \mathcal{F}_1$.

**Remark (Degrees-of-freedom heuristic).** A single-step form $D - \mathbf{u}\mathbf{v}^\top$ has roughly $D + 2D - 1 = 3D - 1$ effective degrees of freedom (accounting for the scale redundancy $(\mathbf{u}, \mathbf{v}) \mapsto (\alpha\mathbf{u}, \alpha^{-1}\mathbf{v})$). A two-step product introduces (at least) twice as many parameters, suggesting a strictly larger family. However, since the parameterization is not injective, we rely on the explicit separation in Proposition 1 (rather than on dimensional counting) to establish strict inclusion.

**Conclusion 2:** There exists $A \in \mathcal{F}_M \setminus \mathcal{F}_1$ for any $M \geq 2$ and $D \geq 3$ (e.g., $A^\star$ above), hence $\mathcal{F}_1 \subsetneq \mathcal{F}_M$.

### A.3 Rank Upper Bound and Expansion for the Injection Term $B(\mathbf{x}_t)$

**Theoretical significance:** Not only does the expressiveness of the state transition $A$ increase, but the capacity of the injection channel $B$ expands in tandem, jointly enhancing the affine transformation capability.

By definition, the injection matrix $B(\mathbf{x}_t) = \sum_{j=0}^{M-1} P_{t,j} R_{t,j}$, where $\mathrm{rank}(R_{t,j}) = 1$.

**Proposition 2:** For fixed input $\mathbf{x}_t$, the rank of $B(\mathbf{x}_t)$ satisfies

$$\mathrm{rank}(B(\mathbf{x}_t)) \leq M. \tag{25}$$

Equality holds when the image spaces of $\{P_{t,j} R_{t,j}\}_j$ are linearly independent.

**Proof:** By the rank inequality for matrix products, $\mathrm{rank}(P_{t,j} R_{t,j}) \leq \min\{\mathrm{rank}(P_{t,j}), \mathrm{rank}(R_{t,j})\} = 1$. Using subadditivity of rank,

$$\mathrm{rank}(B(\mathbf{x}_t)) = \mathrm{rank}\left(\sum_{j=0}^{M-1} P_{t,j} R_{t,j}\right) \leq \sum_{j=0}^{M-1} \mathrm{rank}(P_{t,j} R_{t,j}) \leq M. \tag{26}$$

**Conclusion 3:** $\mathrm{rank}(B(\mathbf{x}_t)) \leq M$, and the upper bound is tight.

## A.4 Spectral Properties and Stability

**Theoretical significance:** Enhanced expressiveness does not come at the expense of numerical stability. The diagonal factors control spectral contraction, while rank-1 terms act as bounded perturbations that contribute expressiveness.

**Lemma 3:** Let $\widetilde{D} = \prod_{j=M-1}^{0} D_{t,j}$. Owing to commutativity of diagonal matrices, the spectral radius $\rho(\cdot)$ satisfies

$$\rho(\widetilde{D}) \le \prod_{j=0}^{M-1} \rho(D_{t,j}). \tag{27}$$

**Proof:** Write $D_{t,j} = \mathrm{diag}(d_{t,j}^{(1)}, \ldots, d_{t,j}^{(D)})$. Then $\widetilde{D} = \mathrm{diag}(\prod_j d_{t,j}^{(1)}, \ldots, \prod_j d_{t,j}^{(D)})$. Hence

$$\rho(\widetilde{D}) = \max_i \left| \prod_j d_{t,j}^{(i)} \right| = \max_i \prod_j \left| d_{t,j}^{(i)} \right| \le \prod_j \left( \max_k \left| d_{t,j}^{(k)} \right| \right) = \prod_j \rho(D_{t,j}). \tag{28}$$

**Lemma 4:** The norm of the difference between $A(\mathbf{x}_t)$ and its diagonal-dominant part $\widetilde{D}$ is bounded. Using the telescoping identity:

$$\prod_{j=0}^{M-1} X_j - \prod_{j=0}^{M-1} Y_j = \sum_{m=0}^{M-1} \left( \prod_{j=0}^{m-1} X_j \right) (X_m - Y_m) \left( \prod_{j=m+1}^{M-1} Y_j \right), \tag{29}$$

and setting $X_j = A_{t,j}$ and $Y_j = D_{t,j}$, we have $X_m - Y_m = -\mathbf{u}_{t,m}\mathbf{v}_{t,m}^\top$. Substituting and applying norm inequalities yields:

$$\left\| A(\mathbf{x}_t) - \prod_j D_{t,j} \right\|_2 \le \sum_{m=0}^{M-1} \left( \prod_{j=0}^{m-1} \|A_{t,j}\|_2 \right) \|\mathbf{u}_{t,m}\|_2 \|\mathbf{v}_{t,m}\|_2 \left( \prod_{j=m+1}^{M-1} \|D_{t,j}\|_2 \right), \tag{30}$$

so the bound depends solely on the norms of sub-step parameters.

**Conclusion 4:** The spectral radius is controlled stepwise by the diagonal factors, while the rank-1 terms serve as bounded perturbations.

## A.5 Strict Expansion of the Affine Transformation Family

**Theoretical significance:** Collecting the above results, we provide a unified functional-family statement of the expressiveness gains afforded by multi-substep updates.

Define the family of affine maps:

$$\mathcal{G}_M := \{ s \mapsto A(\mathbf{x})s + B(\mathbf{x}) : A(\mathbf{x}) \in \mathcal{F}_M, \ B(\mathbf{x}) \text{ satisfies Proposition 2} \}. \tag{31}$$

**Theorem 2:** For $M \ge 2$ and dimension $D \ge 2$, the single-substep affine family $\mathcal{G}_1$ is a strict subset of the multi-substep family $\mathcal{G}_M$:

$$\mathcal{G}_1 \subsetneq \mathcal{G}_M. \tag{32}$$

**Proof:** This follows directly from the preceding results. By Theorem 1, the linear part expands strictly, $\mathcal{F}_1 \subsetneq \mathcal{F}_M$. By Proposition 2, the rank upper bound of the affine injection increases from 1 to $M$. The simultaneous enhancement of both parts ensures a strict expansion of the overall function family.

**Conclusion 5:** $\mathcal{G}_1 \subsetneq \mathcal{G}_M$—the intra-layer expressive power of RWKV-Product increases strictly with the number of sub-steps $M$.

# B  CONSTRUCTIVE PROOF OF CHAIN-BLOCK SOLVING MQAR FOR ARBITRARY SEQUENCE LENGTHS

This appendix proves that a single Chain-Block (RWKV-Product + SGSA) can solve the Multi-Query Associative Recall (MQAR) task (Arora et al., 2024) for arbitrarily long sequences under idealized assumptions. The purpose is to justify the main-text intuition: RWKV-Product performs lossless "write/overwrite" in latent state, while SGSA performs sparse "pointer-read" over local windows and periodic anchors.

## B.1  PROBLEM SETUP AND OBJECTIVE

**MQAR Task (Latest-Write-Wins).** We observe a length-$T$ sequence where some positions are *writes* of key-value pairs $(k_t, v_t)$ and some positions are *queries* with $q_t \in \mathcal{K}$. For each query at time $t$, the correct answer is the value associated with the *most recent* write of the same key:

$$\text{Ans}(t) = v_{\tau^\star}, \qquad \tau^\star = \max\{\tau \le t : k_\tau = q_t\}.$$

This is the Latest-Write-Wins (LWW) semantics.

**Chain-Block.** A Chain-Block consists of: (1) an RWKV-Product state layer that updates a latent state and emits per-token representations; and (2) a State-Guided Sparse Attention (SGSA) layer that performs causal sparse attention over (i) a local window of size $W$ and (ii) a set of periodic anchor positions with stride $G$.

**Goal.** We show there exists a parameter setting such that a single Chain-Block outputs the correct $\text{Ans}(t)$ at every query position $t$, for any sequence length $T$, while respecting LWW.

## B.2  IDEALIZED ASSUMPTIONS (EXISTENCE PROOF)

We assume:

1. **Finite key universe and sufficient dimension.** The key set $\mathcal{K}$ is finite, and the model has enough hidden dimension to allocate a dedicated "associative memory" subspace.

2. **Orthogonal key encoding.** There exists an encoding map $\phi : \mathcal{K} \to \mathbb{R}^{d_k}$ such that

$$\langle \phi(k), \phi(k') \rangle = \delta_{k,k'}.$$

3. **Injective value encoding.** There exists an encoding $\psi : \mathcal{V} \to \mathbb{R}^{d_v}$ that is injective (e.g., one-hot or any separable embedding).

4. **Sufficient numerical precision.** We can make attention "hard" by scaling logits and we can treat linear updates as exact up to an arbitrarily small error.

5. **RWKV-Product expressivity for low-rank affine state updates.** RWKV-Product can realize (up to small error) the rank-1 projector-and-inject updates described below on a designated memory subspace, by choosing appropriate diagonal decay and rank-1 terms (consistent with its "diagonal + low-rank" structure).

## B.3  MEMORY ENCODING FOR MQAR

We use an explicit associative-memory construction.

**Outer-product notation.** We use $\mathbf{ab}^\top$ for the outer product, producing a matrix in $\mathbb{R}^{d_k \times d_v}$.

**Periodic anchors.** Define the anchor set

$$\mathcal{A} = \{t \in \{1, \ldots, T\} : t \bmod G = G - 1\}.$$

SGSA will allow each query $t$ to attend to positions in

$$\mathcal{D}(t) = \{t' : \max(1, t - W) \le t' \le t\} \cup \{a \in \mathcal{A} : a \le t\}.$$

**Two-level memory: segment buffer + global memory.** We maintain two associative memories:

- **Segment (recent) memory** $(P_t^{\text{seg}}, M_t^{\text{seg}})$ that stores writes *since the most recent anchor*.
- **Global memory** $(P_t^{\text{glob}}, M_t^{\text{glob}})$ that stores all writes *up to time $t$*.

Each memory consists of a **presence** vector $P \in \mathbb{R}^{d_k}$ and a **value** matrix $M \in \mathbb{R}^{d_k \times d_v}$:

$$P_t(\cdot) \approx \sum_{k \in \mathcal{K}} \mathbf{1}\{k \text{ has appeared}\}\, \phi(k), \qquad M_t \approx \sum_{k \in \mathcal{K}} \phi(k)\, \psi(v_k(t))^\top,$$

where $v_k(t)$ denotes the latest written value of key $k$ up to time $t$.

**Correct readout (no vec/Kronecker required).** If the above invariant holds, then for any query key $q \in \mathcal{K}$,

$$M_t^\top\, \phi(q) \;=\; \psi(v_q(t)), \tag{33}$$

i.e., multiplying by $\phi(q)$ selects the corresponding value embedding (because $\phi(\cdot)$ are orthonormal).

### B.4 RWKV-PRODUCT LAYER: LWW WRITE/OVERWRITE AND PERSISTENCE

We now specify update rules that enforce LWW exactly, and argue RWKV-Product can implement them on its memory subspace.

**Rank-1 overwrite for LWW.** Let $e_t := \phi(k_t) \in \mathbb{R}^{d_k}$ and $u_t := \psi(v_t) \in \mathbb{R}^{d_v}$ for a write token at time $t$. Define the *project-and-inject* update:

$$M_t \;=\; (I - e_t e_t^\top)\, M_{t-1} \;+\; e_t u_t^\top. \tag{34}$$

This overwrites the row corresponding to $k_t$ with $u_t$ while leaving all other keys unchanged:

$$M_t^\top \phi(k_t) = u_t, \qquad M_t^\top \phi(k') = M_{t-1}^\top \phi(k') \text{ for } k' \neq k_t.$$

Likewise, presence can be maintained by

$$P_t \;=\; (I - e_t e_t^\top)\, P_{t-1} \;+\; e_t, \tag{35}$$

which sets the coefficient of key $k_t$ in $P_t$ to 1 (in the orthonormal basis).

**Global and segment updates.** For a **write** at time $t$:

$$M_t^{\text{glob}} = (I - e_t e_t^\top) M_{t-1}^{\text{glob}} + e_t u_t^\top, \quad P_t^{\text{glob}} = (I - e_t e_t^\top) P_{t-1}^{\text{glob}} + e_t,$$

and similarly for the segment memory:

$$M_t^{\text{seg}} = (I - e_t e_t^\top) M_{t-1}^{\text{seg}} + e_t u_t^\top, \quad P_t^{\text{seg}} = (I - e_t e_t^\top) P_{t-1}^{\text{seg}} + e_t.$$

For a **non-write** token, we keep both memories unchanged (identity transition).

At an **anchor** position $t \in \mathcal{A}$, after producing the token representation (see below), we *reset* the segment buffer:

$$M_t^{\text{seg}} = 0, \qquad P_t^{\text{seg}} = 0. \tag{36}$$

This ensures the segment buffer always represents writes after the last anchor.

RWKV-Product updates its state via an affine map with a "diagonal + low-rank" transition and low-rank injection. On a designated memory subspace, one can choose: (i) diagonal factors close to 1 (to preserve memory), (ii) rank-1 terms proportional to $e_t e_t^\top$ (to realize $(I - e_t e_t^\top)$), and (iii) injection terms proportional to $e_t u_t^\top$ (to realize the overwrite). Anchor resets in (36) can be implemented by conditioning on a binary anchor-indicator feature $\mathbf{1}_{\{t \in \mathcal{A}\}}$ and applying a diagonal contraction to the segment channels. Since RWKV-Product supports multi-substep low-rank updates, these operations can be composed within a single time step on disjoint channel blocks.

### B.5 WHAT RWKV-PRODUCT EMITS TO SGSA

Let $y_t$ denote the RWKV-Product output at position $t$ (the input to SGSA). We construct $y_t$ to expose:

- **Segment summary (always visible locally):** a linearized payload of $(P_t^{\text{seg}}, M_t^{\text{seg}})$.
- **Global checkpoint (only at anchors):** if $t \in \mathcal{A}$, additionally expose a payload of $(P_t^{\text{glob}}, M_t^{\text{glob}})$; otherwise this part is zeroed.
- **Anchor indicator:** a bit $\mathbf{1}_{\{t \in \mathcal{A}\}}$.

Concretely, we may write (schematically)

$$y_t = \Big[ \underbrace{P_t^{\text{seg}} \oplus \text{vec}(M_t^{\text{seg}})}_{\text{segment payload}} \oplus \mathbf{1}_{\{t \in \mathcal{A}\}} \cdot \underbrace{\big(P_t^{\text{glob}} \oplus \text{vec}(M_t^{\text{glob}})\big)}_{\text{anchor payload}} \oplus (\text{other channels})\Big].$$

This matches the intended "recent write in local window + far history at sparse anchors" decomposition.

### B.6 SGSA LAYER: SPARSE POINTER-READ RETRIEVES THE CORRECT VALUE

We now show SGSA can recover the correct value for each query using only the sparse domain $\mathcal{D}(t)$.

**Attention domain.** For a query position $t$, SGSA attends over

$$\mathcal{D}(t) = \{t - W, \dots, t\} \cup \mathcal{A}_{\leq t}.$$

**Keys: presence-based addressability.** Define the query vector (in the key-embedding space) as

$$Q_t = \alpha\, \phi(q_t), \qquad \alpha > 0.$$

For each visible position $\text{pos} \in \mathcal{D}(t)$, define its addressing key as

$$K_{\text{pos}} = \begin{cases} P_{\text{pos}}^{\text{seg}}, & \text{pos} \notin \mathcal{A}, \\ P_{\text{pos}}^{\text{glob}}, & \text{pos} \in \mathcal{A}. \end{cases}$$

Then, by orthogonality,

$$\langle Q_t, K_{\text{pos}} \rangle = \alpha \cdot \langle \phi(q_t), K_{\text{pos}} \rangle \in \{0, \alpha\},$$

indicating whether the queried key $q_t$ is present in that memory snapshot.

**Recency bias enforces LWW selection.** Let the attention score be

$$\text{score}(t, \text{pos}) = \langle Q_t, K_{\text{pos}} \rangle + \rho(\text{pos} - t), \qquad (37)$$

where $\rho(\Delta)$ is any *strictly increasing* function of $\Delta \leq 0$ (i.e., more recent positions get larger bias). Under the causal mask, $\text{pos} \leq t$ always holds.

In the limit $\alpha \to \infty$, the softmax concentrates on the most recent visible position where the key is present:

$$\text{pos}^\star(t) = \arg \max_{\text{pos} \in \mathcal{D}(t)} \text{score}(t, \text{pos}) = \max\{\text{pos} \in \mathcal{D}(t) : \langle \phi(q_t), K_{\text{pos}} \rangle = 1\}.$$

**Values: return the selected memory payload.** Let SGSA return (approximately) the value payload of the selected position:

$$\widetilde{m}_t \approx \begin{cases} \text{vec}(M_t^{\text{seg}}), & \text{pos}^\star(t) = t \text{ (segment case)}, \\ \text{vec}(M_{\text{pos}^\star(t)}^{\text{glob}}), & \text{pos}^\star(t) \in \mathcal{A} \text{ (anchor case)}. \end{cases}$$

(Using self-attention, $\text{pos}^\star(t) = t$ is allowed.)

**Final readout.** From the selected matrix $M_{\mathrm{sel}}$ (segment or global), we obtain the value embedding by

$$\widehat{\psi}(v) \;=\; M_{\mathrm{sel}}^{\top}\phi(q_t),$$

which equals the correct $\psi(\mathrm{Ans}(t))$ by the LWW invariants of (34). This multiplication is a standard associative-memory read; it can be implemented by a fixed readout head that consumes both the query representation (which contains $\phi(q_t)$) and the retrieved payload (which contains $M_{\mathrm{sel}}$), e.g., via a bilinear form on the designated subspace.

### B.7 Correctness Sketch

We summarize why the above construction solves MQAR for any $T$.

**Case 1: the latest write of $q_t$ occurs after the last anchor.** Then $q_t$ is present in the segment buffer at time $t$, i.e., $\langle\phi(q_t), P_t^{\mathrm{seg}}\rangle = 1$, so position $t$ itself is a valid match. By the strictly increasing recency bias $\rho$, $\mathrm{pos}^{\star}(t) = t$. SGSA retrieves $M_t^{\mathrm{seg}}$, and by (33) outputs the value of the latest write within the segment, which equals the latest write overall.

**Case 2: the latest write of $q_t$ occurs at or before the last anchor.** Then $\langle\phi(q_t), P_t^{\mathrm{seg}}\rangle = 0$ (it is absent from the segment buffer), but $q_t$ is present in the global memory checkpoint at the most recent anchor $a \leq t$. Thus $\mathrm{pos}^{\star}(t) = a$, and SGSA retrieves $M_a^{\mathrm{glob}}$ which encodes the latest value up to $a$. Since there is no later write after $a$ (by assumption of this case), this equals the correct LWW answer.

**Therefore,** each query retrieves $\psi(\mathrm{Ans}(t))$ exactly (under the idealized assumptions). Since anchors exist periodically, the argument does not depend on $T$.

## C Turing Completeness of ChainGPT

This appendix proves that the proposed *recurrent depth* formulation of ChainGPT is, in principle, computationally universal.

### C.1 Setup and Main Result

Recall the recurrent depth formulation in the main text. Let the input token sequence be embedded and processed by a bottom module to obtain a *static basis* (read-only context)

$$H_0 = \mathrm{Bottom}(\mathrm{Embed}(X)) \in \mathbb{R}^{S\times d}, \tag{38}$$

which remains fixed during recursion. Let $X_t \in \mathbb{R}^{S\times d}$ denote the *writable* latent state at recurrent iteration $t$, updated by a fixed-parameter map

$$X_{t+1} = F_\theta(X_t, H_0), \tag{39}$$

where $\theta$ is shared across all iterations (a single "interpreter step" repeatedly applied).

**Idealized conditions.** We make the following standard assumptions, which are common in universality proofs for neural architectures:

1. **Unbounded workspace and time:** the sequence length $S$ (workspace size) and the number of recurrent iterations can be arbitrarily large.

2. **Arbitrary-precision real computation:** internal activations support arbitrarily precise real values. Equivalently, we may assume that sufficiently sharp winner-take-all selection (hard-max limit) and saturated gates can be implemented to any desired tolerance.

3. **Structured primitives available in a fixed-depth block:** the architecture provides (i) residual connections, (ii) elementwise gating/multiplicative interactions (e.g., gated-MLP / RWKV-style gates) enabling masked overwrite, and (iii) position-adjacent access operators (e.g., `time_shift` or an equivalent fixed Toeplitz/conv operator).

4. **Sparse attention compatibility:** for the theoretical construction, we store the program on *global anchor* positions so that it is always visible under SGSA (or, as a degenerate special case, we set the anchor stride $G = 1$ so that all program slots are anchors). This ensures random access to program instructions within the sparse retrieval domain.

Under these assumptions, we prove computational universality by simulation of a two-counter Minsky machine (2CM), which is known to be equivalent in power to a Turing machine.

**Theorem C.1 (Turing Completeness).** *Under the above idealized conditions, ChainGPT with recurrent depth updates* (39) *can simulate an arbitrary Turing machine; hence it is Turing complete in principle.*

We give a constructive simulation of one step of a 2CM by a single application of $F_\theta$. Since any Turing machine can be compiled into a 2CM program, repeated application of $F_\theta$ reproduces the computation trace.

## C.2 STATE LAYOUT AND ENCODING

We interpret the sequence axis as a 1D workspace of length $S$ (a "tape"), where each position stores a $d$-dimensional vector. We reserve contiguous segments of positions for different roles:

$$\{1, \ldots, S\} = \mathcal{P} \cup \mathcal{R}_1 \cup \mathcal{R}_2 \cup \mathcal{U}, \tag{40}$$

where:

- $\mathcal{P}$ is a **program segment** containing instruction slots (read-only, stored in $H_0$);
- $\mathcal{R}_1, \mathcal{R}_2$ are **register segments** (writable, stored in $X_t$) for counters $R_1, R_2$;
- $\mathcal{U}$ is unused padding (all zeros), allowing the workspace to expand.

**Program storage on anchors (SGSA compatibility).** Let the SGSA anchor set be

$$\mathcal{A} = \{i \in \{1, \ldots, S\} : i \bmod G = G - 1\}. \tag{41}$$

For the proof, we place each instruction slot at an anchor position: $\mathcal{P} \subseteq \mathcal{A}$ (or simply take $G = 1$). Thus, any causal SGSA query position occurring after $\mathcal{P}$ can attend to all program slots through the anchor links.

**Instruction representation.** A 2CM program is a finite list of instructions indexed by $m \in \{1, \ldots, M_{\text{prog}}\}$, each of the form

$$(\text{op}, r, \text{target}, \text{next}), \quad \text{op} \in \{\text{INC}, \text{DECJZ}\}, \ r \in \{1, 2\}.$$

We store each instruction in $H_0$ at one program position $p(m) \in \mathcal{P}$ using fixed channels:

- an **op code** indicator (INC vs. DECJZ),
- a **register selector** indicator ($r = 1$ or $r = 2$),
- two **one-hot masks** over $\mathcal{P}$: $\text{TargetMask}(m) \in \{0, 1\}^{|\mathcal{P}|}$ and $\text{NextMask}(m) \in \{0, 1\}^{|\mathcal{P}|}$, embedded as channels aligned with program positions.

This "mask-in-program" representation avoids explicit arithmetic on program indices and makes jumps purely mask-based.

**Program counter (PC) encoding.** We represent the current instruction pointer as a one-hot mask over program slots, stored in a dedicated channel of the writable state:

$$\text{PC}_t(i) \in \{0, 1\}, \ i \in \mathcal{P}, \qquad \sum_{i \in \mathcal{P}} \text{PC}_t(i) = 1. \tag{42}$$

Intuitively, exactly one program slot is "active" at each step.

**Register encoding (unary with delimiter).** For each register $R_r$ ($r \in \{1, 2\}$), we use a unary encoding within segment $\mathcal{R}_r$:

$$R_r = n \quad \Longleftrightarrow \quad [\underbrace{1, 1, \ldots, 1}_{n \text{ ones}} \mid 0, 0, \ldots], \tag{43}$$

where the delimiter "|" is represented by a one-hot **delimiter mask channel**

$$\text{Delim}_t^{(r)}(i) \in \{0, 1\}, \ i \in \mathcal{R}_r, \qquad \sum_{i \in \mathcal{R}_r} \text{Delim}_t^{(r)}(i) = 1. \tag{44}$$

The tape also contains a **data channel** holding the cell value (approximately 0 or 1). The delimiter position marks the boundary between ones and zeros.

## C.3 Primitive Operations Realizable by One Fixed-Depth Step

We now outline how one application of $F_\theta$ can implement the required discrete operations using a fixed-depth composition of Chain-Blocks (state update + SGSA + gated-MLP), under the idealized assumptions.

### C.3.1 Pointer-driven instruction fetch under SGSA

Because program slots lie on anchors ($\mathcal{P} \subseteq \mathcal{A}$), SGSA can retrieve from all instruction positions via sparse anchor links.

Let $u(i) \in \mathbb{R}^d$ denote the instruction embedding stored at program slot $i \in \mathcal{P}$ (as part of $H_0$). We define an attention head (or an equivalent linear aggregation) that computes an instruction summary

$$\text{Instr}_t = \sum_{i \in \mathcal{P}} \text{PC}_t(i)\, u(i), \tag{45}$$

which equals the active instruction embedding since $\text{PC}_t$ is one-hot.

**Implementation note.** Equation (45) can be realized by SGSA by setting keys to include the $\text{PC}_t(i)$ channel (or using it to bias logits) and using a sufficiently sharp selection temperature. Under arbitrary precision, the softmax can approximate hard selection to any desired tolerance; we additionally apply saturated gates to keep $\text{PC}_t$ binary-stable (see §C.5).

### C.3.2 Adjacency access via shift operators

We assume access to fixed shift operators along the sequence axis for designated mask channels:

$$\text{ShiftRight}(z)(i) = z(i-1), \qquad \text{ShiftLeft}(z)(i) = z(i+1). \tag{46}$$

Such operators can be implemented by RWKV-style `time_shift` or an equivalent fixed convolution/Toeplitz linear map.

These shifts allow each cell to locally detect whether it is the delimiter position or a neighbor of the delimiter, enabling local rule-based updates without global write operations.

### C.3.3 Masked overwrite (constant-support writeback)

A key requirement is to modify only a constant number of tape locations per instruction (in the sense of *net* change), while leaving the remaining cells unchanged.

Given a (possibly multi-hot) mask $m(i) \in \{0, 1\}$ supported on a small index set $I$, a gated-MLP can implement masked overwrite:

$$X'(i) = X(i) + m(i) \odot \big(V(i) - X(i)\big), \tag{47}$$

where $V(i)$ is the desired writeback value at masked positions and $\odot$ denotes elementwise multiplication provided by gating. When $m$ is supported on $I$, only positions in $I$ change.

In our construction, $m(i)$ will be formed from delimiter masks and their shifted versions, so $|I|$ is a constant (e.g., $\{b\}$ or $\{b, b \pm 1\}$).

### C.3.4 Zero test for unary register

Let $\mathcal{R}_r$ be the register-$r$ segment. Let $b$ be the delimiter position, i.e., $\text{Delim}_t^{(r)}(b) = 1$. The register is zero iff the delimiter is at the left boundary of the segment, equivalently there is no "1" immediately before it inside the segment.

Using shifts, we can form a *pre-delimiter mask*:

$$\text{PreDelim}_t^{(r)} = \text{ShiftLeft}\big(\text{Delim}_t^{(r)}\big), \tag{48}$$

and then read the data channel at that location. A piecewise-linear threshold (hard-sigmoid in the idealized setting) yields a binary flag:

$$\text{NonZero}_t^{(r)} \in \{0, 1\}, \qquad \text{FLAG}_t = 1 - \text{NonZero}_t^{(r)}. \tag{49}$$

For DECJZ, $\text{FLAG}_t = 1$ indicates the jump-on-zero branch.

## C.4 SIMULATING ONE STEP OF A 2CM

We now describe how $F_\theta$ implements the two instruction types.

### C.4.1 INC

Suppose the fetched instruction is $(\mathrm{INC}, r, \mathrm{target}, \mathrm{next})$. Let $\mathrm{Delim}_t^{(r)}$ denote the delimiter mask for register $r$. Define

$$m_{\mathrm{inc}} = \mathrm{Delim}_t^{(r)} + \mathrm{ShiftRight}\big(\mathrm{Delim}_t^{(r)}\big), \tag{50}$$

which is supported on $\{b, b+1\}$. Using (47), we overwrite:

- at $b$: write data-value 1 and clear delimiter;
- at $b+1$: set delimiter and keep data-value 0 (or leave as 0).

All other positions remain unchanged.

The program counter advances deterministically:

$$\mathrm{PC}_{t+1} = \mathrm{NextMask}(\mathrm{Instr}_t), \tag{51}$$

where $\mathrm{NextMask}(\mathrm{Instr}_t)$ is decoded from the fetched instruction embedding (a one-hot mask over $\mathcal{P}$).

### C.4.2 DECJZ

Suppose the fetched instruction is $(\mathrm{DECJZ}, r, \mathrm{target}, \mathrm{next})$.

First compute the zero-test flag $\mathrm{FLAG}_t \in \{0, 1\}$ as in §C.3.4.

- If $\mathrm{FLAG}_t = 1$ (zero case), we leave the register unchanged and jump:

$$\mathrm{PC}_{t+1} = \mathrm{TargetMask}(\mathrm{Instr}_t). \tag{52}$$

- If $\mathrm{FLAG}_t = 0$ (nonzero case), we decrement by moving the delimiter left by one: define

$$m_{\mathrm{dec}} = \mathrm{Delim}_t^{(r)} + \mathrm{ShiftLeft}\big(\mathrm{Delim}_t^{(r)}\big), \tag{53}$$

  supported on $\{b-1, b\}$, and overwrite:

    - at $b-1$: set delimiter;
    - at $b$: clear delimiter and write data-value 0.

  Then advance:

$$\mathrm{PC}_{t+1} = \mathrm{NextMask}(\mathrm{Instr}_t). \tag{54}$$

## C.5 BINARY STABILITY AND ERROR CONTROL

Because the construction relies on near-discrete masks (PC and delimiters), we include a stability mechanism to prevent small numerical leakage from accumulating over arbitrarily many recurrent iterations.

After each step, we apply a *bistable contraction* (e.g., hard-sigmoid with thresholds) on the dedicated mask channels:

$$\Pi(z) = \begin{cases} 0, & z \leq \frac{1}{3}, \\ 3z - 1, & \frac{1}{3} < z < \frac{2}{3}, \\ 1, & z \geq \frac{2}{3}, \end{cases} \tag{55}$$

ensuring that PC and delimiter masks remain within an $\varepsilon$-neighborhood of $\{0, 1\}$, and the one-hot constraints remain satisfied up to an arbitrarily small tolerance (guaranteed by the arbitrary-precision assumption).

### C.6 CORRECTNESS OF THE SIMULATION

Let $\text{Step}_{2CM}$ denote the true one-step transition function of the 2CM on configurations encoded as $(\text{PC}, \text{Delim}^{(1)}, \text{Delim}^{(2)}, \text{data})$.

**Lemma C.1 (Single-step correctness).** *There exist fixed parameters $\theta$ such that for any valid encoded 2CM configuration $X_t$, one application of $F_\theta$ satisfies*

$$F_\theta(X_t, H_0) = X_{t+1} = \text{Encode}\big(\text{Step}_{2CM}(\text{Decode}(X_t))\big), \tag{56}$$

*and the net modification of the tape is supported on a constant-size set of positions (those adjacent to the active delimiter), while all other positions are preserved by residual identity paths.*

Instruction fetch (45) selects exactly the active instruction. The INC/DECJZ updates are implemented by masked overwrite (47) with masks supported on $\{b, b \pm 1\}$. PC updates (51)–(54) are mask-based, thus remain one-hot after applying $\Pi(\cdot)$. Residual connections preserve unmodified positions.

**Lemma C.2 (Multi-step correctness).** *By induction on $t$, repeated application of $F_\theta$ produces a sequence $\{X_t\}$ that matches the 2CM execution trace step-by-step.*

### C.7 FROM 2CM TO TURING MACHINE

It is a classical result that a two-counter Minsky machine can simulate an arbitrary Turing machine (and vice versa). Therefore, for any Turing machine $M$, there exists a 2CM program $P$ that simulates $M$. We encode $P$ into the read-only basis $H_0$ on the program segment $\mathcal{P}$, and encode the initial tape/state of $M$ into the writable register segments $\mathcal{R}_1, \mathcal{R}_2$ (and auxiliary padding $\mathcal{U}$).

If $M$ halts, the corresponding 2CM reaches a halting configuration. We can represent halting by a dedicated HALT instruction that forces PC into an absorbing slot and/or sets a halting flag channel to 1, after which $F_\theta$ becomes the identity map on the writable state. The output module can then read out the final configuration (e.g., via the logits at a designated readout position).

Combining Lemma C.2 with the 2CM–TM equivalence yields Theorem C.1.

## D ALGORITHMIC IMPLEMENTATION DETAILS

For clarity, this section provides the core algorithmic implementations of ChainGPT.

### D.1 RWKV-PRODUCT

Given an input sequence $H = \{h_t\}_{t=1}^T \in \mathbb{R}^{B \times T \times C}$ and the number of sub-steps $M$. Let the state matrix sequence be $S = \{s_1, s_2, \ldots, s_T\}$.

Initialization:
$$s_0 = 0$$

For $t = 1$ to $T$:

$$\tilde{h}_t = \begin{cases} 0, & t = 1 \\ h_{t-1}, & t > 1 \end{cases}$$

$$\Delta h_t = \tilde{h}_t - h_t$$
$$x_t^r = h_t + \Delta h_t \odot x^r$$
$$x_t^w = h_t + \Delta h_t \odot x^w$$
$$x_t^k = h_t + \Delta h_t \odot x^k$$
$$x_t^v = h_t + \Delta h_t \odot x^v$$
$$x_t^\alpha = h_t + \Delta h_t \odot x^\alpha$$
$$x_t^g = h_t + \Delta h_t \odot x^g$$

$$R_t = W_r x_t^r$$

$$W_t = -\text{softplus}(-(w_0 + \tanh(x_t^w w_1)w_2)) - 0.5$$

$$K_t^0 = W_k x_t^k$$

$$V_t^0 = W_v x_t^v$$

For $j = 0$ to $M - 1$:

$$r_{t,j} = \begin{cases} R_t, & j = M - 1 \\ 0, & \text{otherwise} \end{cases}$$

$$w_{t,j} = \begin{cases} W_t, & j = 0 \\ -\infty, & \text{otherwise} \end{cases}$$

$$\Delta K_{t,j}^b = x_t^k A_j^b B_j^b$$

$$\Delta K_{t,j}^c = x_t^k A_j^c B_j^c$$

$$K_{t,j}^b = K_t^0 + \Delta K_{t,j}^b$$

$$K_{t,j}^c = K_t^0 + \Delta K_{t,j}^c$$

$$\beta_{t,j}^b = \sigma(\alpha_{0,j}^b + (x_t^\alpha \alpha_{1,j}^b)\alpha_{2,j}^b)$$

$$\beta_{t,j}^c = \sigma(\alpha_{0,j}^c + (x_t^\alpha \alpha_{1,j}^c)\alpha_{2,j}^c)$$

$$\hat{K}_{t,j}^c = K_{t,j}^c \odot [1 + (\beta_{t,j}^c - 1) \cdot k^{ac}]$$

$$b_{t,j}^- = -\frac{K_{t,j}^b}{\|K_{t,j}^b\|}$$

$$b_{t,j}^+ = \frac{K_{t,j}^b}{\|K_{t,j}^b\|} \odot \beta_{t,j}^b$$

$$V_{t,j} = V_t^0 + x_t^v A_{j-1}^v B_{j-1}^v$$

Call the CUDA implementation:

$$X_{\text{inter}} = \text{RUN\_CUDA\_RWKV7g}(r, w, \hat{k}^c, v, b^-, b^+, H_{\text{head}})$$

$$X_{\text{att}} = \text{reshape}(X_{\text{inter}}, B, T, M, C)[:, :, -1, :]$$

$$X_{\text{gn}} = \text{GroupNorm}(X_{\text{att}})$$

$$\text{term2}_t = \sum_{h=1}^{H} (r_{t,M-1}^h \cdot k_{t,M-1}^h \cdot r_k^h)v_{t,M-1}^h$$

$$X' = X_{\text{gn}} + \text{term2}_t$$

$$g_t = \sigma(x_t^g g_1)g_2$$

$$s_t = W_o(X' \odot g_t)$$

## D.2 STATE-GUIDED SPARSE ATTENTION

Given a state sequence $X \in \mathbb{R}^{B \times S \times C}$, window size $W$, and global interval $G$.

$$Q = XW_q, \quad K = XW_k, \quad V = XW_v$$
$$Q, K, V \in \mathbb{R}^{B \times H \times S \times D}$$

For $i = 0$ to $d/2 - 1$:

$$\theta_t^{(i)} = t/10000^{2i/d}$$
$$f = i/(d/2)$$
$$\beta(f) = \beta_{\text{slow}} + (\beta_{\text{fast}} - \beta_{\text{slow}})f$$
$$n_s = 1 + \frac{\alpha - 1}{\beta(f)}$$
$$\tilde{\theta}_t^{(i)} = \theta_t^{(i)}/n_s$$

Apply $\tilde{\theta}_t^{(i)}$ to rotate-encode $q_{2i}, q_{2i+1}$ and $k_{2i}, k_{2i+1}$.

$$Q' = \text{RoPE}(Q), \quad K' = \text{RoPE}(K)$$

Construct a sparse mask $M_{ij}$ for $i, j = 1$ to $S$:

$$M_{ij} = \begin{cases} 0, & i - j < W \text{ and } j \leq i \\ 0, & j \bmod G = G - 1 \text{ and } j \leq i \\ -\infty, & \text{otherwise} \end{cases}$$

If num_kv_heads $< H$, expand $K', V'$ to match the head dimension.

$$\text{Attention}(Q', K', V'; M) = \text{softmax}\left(\frac{Q'K'^\top}{\sqrt{d}} + M\right)V'$$

$$\text{Out} = \text{concat heads} \cdot W_o$$

## D.3 RECURRENT DEPTH APPROACH

Let the input sequence be $X \in \mathbb{R}^{B \times S}$, and the hidden dimension be $d$.

**Embedding and Bottom-Level:**

$$H = \text{Embed}(X) \in \mathbb{R}^{B \times S \times d}, \qquad H_0 = \text{Bottom}(H) \in \mathbb{R}^{B \times S \times d}.$$

**Looped Iterations:**

$$Z_t = [\,X_t; H_0\,] \in \mathbb{R}^{B \times S \times 2d}, \qquad \tilde{X}_t = Z_t W_a + b_a \in \mathbb{R}^{B \times S \times d}, \qquad X_{t+1} = \mathcal{R}(\tilde{X}_t).$$

**Two-Phase (no-gradient $n_0$ steps + backprop $n_k$ steps) with truncation:**

(No-Grad Phase)      $X_{t+1} = \mathcal{R}([\,X_t; H_0\,]W_a + b_a), \qquad t = 0, \ldots, n_0 - 1,$

(Truncation Boundary)      $\bar{X}_0 = \text{stopgrad}(X_{n_0}),$

(Backprop Phase)      $\bar{X}_{k+1} = \mathcal{C}\Big(\mathcal{R}([\,\bar{X}_k; H_0\,]W_a + b_a)\Big), \qquad k = 0, \ldots, n_k - 1,$

(Output)      $X^\star = \bar{X}_{n_k}.$

**Top Layers and Normalization:**

$$X^{\text{top}} = \text{Top}(X_T), \qquad Y = \text{Norm}(X^{\text{top}}).$$

**Adaptive Inner-Loop Reasoning:** After the $t$-th iteration, obtain the hidden state

$$x_t \in \mathbb{R}^{B \times S \times d}.$$

Decode logits by running only the top blocks:

$$\text{logits}_t = \text{Decode}(x_t) \in \mathbb{R}^{B \times S \times V},$$

where $V$ is the vocabulary size.

Extract the distribution at the last position:

$$\ell_t = \text{logits}_t[:, -1, :] \in \mathbb{R}^{B \times V}.$$

Convert to probabilities:

$$p_t = \text{softmax}(\ell_t) \in \mathbb{R}^{B \times V}.$$

Compute the entropy for each batch element $b$:

$$H_t(b) = -\sum_{i=1}^{V} p_{t,i}(b) \log \left( p_{t,i}(b) + \varepsilon \right), \qquad \varepsilon = 10^{-10}.$$

Define the entropy difference:

$$\Delta H_t(b) = H_{t-k}(b) - H_t(b),$$

where $k$ is the checking interval.

**Early-Stopping Criterion (entropy-diff):**

$$\Delta H_t(b) \leq \tau \quad \Longrightarrow \quad \text{terminate early at step } t,$$

where $\tau$ is a threshold, either a constant or an adaptive function (e.g., $\tau(t) = c/\sqrt{t}$).

## E  COMPARISON OF RECENT RNN MODULES FOR LANGUAGE MODELING (TABLE 6)

## F  EXPERIMENTAL REPRODUCTION GUIDE

The code are provided in the supplementary materials. For convenience, Table 7 lists the main hyperparameters for ChainGPT models at different scales.

For MQAR task reproduction, a wide-range time convolution needs to be applied at the RWKV-Product layer to ensure stability. This is an optimization for training stability and does not affect related conclusions. The experimental setup follows Zoology (Arora et al., 2024).

You can calculate the total number of model parameters using the following method. Let $V$ be the vocabulary size, $D$ the hidden dimension, $L$ the number of decoder layers, with $L_e = \lceil L/2 \rceil$ even-indexed RWKV-v7 layers and $L_o = \lfloor L/2 \rfloor$ odd-indexed GQA layers. Let $D_m$ be the MLP intermediate size, $M$ the micro-step count, $r$ the LoRA rank, $D_{\text{DEC}}, D_{\text{GATE}}, D_{\text{AAA}}$ the low-rank bottlenecks, and $H_q, H_{kv}, d_h$ the query heads, key/value heads, and per-head dimension, with $K = H_{kv} d_h$. Non-layer terms are:

$$P_{\text{embed}} = VD, \quad P_{\text{lm}} = DV, \quad P_{\text{adapter}} = 2D^2 + D, \quad P_{\text{core-norm}} = 2D,$$

and per-layer shared components:

$$P_{\text{2LN}} = 4D, \qquad P_{\text{MLP}} = 3DD_m.$$

Table 6: Comparison of recent RNN modules used for language modeling

| Architecture | State Evolution | FP | DP | HRU | GE |
|---|---|---|---|---|---|
| RWKV-4 | $s_t = e^{-w} \odot s_{t-1} + e^{k_t} \odot v_t$ | × | × | × | × |
| RetNet | $S_t = wS_{t-1} + v_t^T k_t$ | × | × | × | × |
| RWKV-5 | $s_t' = e^{-w} \odot s_{t-1}' + e^{k_t}$ | ✓ | × | × | × |
| Mamba | $S_t = S_{t-1}\text{diag}(w) + v_t^T k_t$ | ✓ | × | × | × |
| RWKV-6 & GLA | $S_t = S_{t-1} \odot \exp(-(w_t^T \mathbf{1}) \odot \exp(A)) + (w_t \odot v_t)^T k_t$ | ✓ | × | × | × |
| HGRN-2 | $S_t = S_{t-1}\text{diag}(w_t) + v_t^T k_t$ | ✓ | × | × | × |
| Mamba-2 | $S_t = S_{t-1}\text{diag}(w_t) + v_t^T(1 - w_t)$ | × | × | × | ✓ |
| TTT[a] | $S_t = w_t S_{t-1} + v_t^T k_t$ | × | × | × | × |
| Gated DeltaNet | $S_t = S_{t-1} \odot (I - a_t^T k_t^2) + (a_t x_t)^T k_t$ | × | × | × | ✓ |
| Titans[a] | $\begin{aligned} S_t &= w_t S_{t-1}(I - a_t k_t^T k_t) + a_t v_t^T k_t \\ M_t &= (1 - \alpha_t)M_{t-1} + S_t \\ S_t &= w_t S_{t-1} - a_t \nabla l(M_{t-1}, k_t, v_t) \end{aligned}$ | × | × | × | ✓ |
| **RWKV-7** | $S_t = S_{t-1}(\text{diag}(w_t) - \hat\kappa_t^T(a_t \odot \hat\kappa_t)) + v_t^T k_t$ | ✓ | × | × | ✓ |
| **DeltaProduct** | $H_i = \left(\prod_{j=1}^{n_h}(I - \beta_{i,j} k_{i,j} k_{i,j}^T)\right) H_{i-1} + \sum_{j=1}^{n_h}\left(\prod_{k=j+1}^{n_h}(I - \beta_{i,k} k_{i,k} k_{i,k}^T)\right)\beta_{i,j} k_{i,j} v_{i,j}^T$ | × | × | ✓ | ✓ |
| **RWKV-Product (ours)** | $\begin{aligned} S_t &= A(x_t)S_{t-1} + B(x_t), \text{ where} \\ A(x_t) &= \prod_{j=0}^{M-1}(\text{diag}(a_{t,j}) - \beta_{b,j} k_{t,j}^{(b)} k_{t,j}^{(b)T}) \\ B(x_t) &= \sum_{j=0}^{M-1}\left(\prod_{k=j+1}^{M-1} A_{t,k}\right)(\beta_{c,j} k_{t,j}^{(c)} v_{t,j}^T) \end{aligned}$ | ✓ | ✓ | ✓ | ✓ |

**FP** (Full-dim Params): all parameters match the model dimensionality, enabling adaptive channel-wise modulation.

**DP** (Decoupled Path): parameters for state transition and information injection are linearly independent, improving expressiveness and convergence.

**HRU** (High-Rank Update): state updates possess a multi-substep high-rank structure, allowing a single-step update to capture complex, diverse state changes.

**GE** (Generalized Eigenvalue): the state update matrix admits a generalized eigenvalue factorization, enabling efficient diagonal-form updates.

Table 7: Main hyperparameters for different model scales

| Parameter | 180M | 0.5B | 1.5B |
|---|---|---|---|
| RWKV head dim | 64 | 64 | 64 |
| GQA head dim | 128 | 128 | 128 |
| Hidden dim | 1024 | 2048 | 2560 |
| KV heads | 2 | 2 | 8 |
| Layers | 8 | 8 | 12 |
| A_low_rank_dim | 64 | 96 | 128 |
| Decay_low_rank_dim | 64 | 96 | 128 |
| Gate_low_rank_dim | 128 | 256 | 320 |
| M_rank_dim | 96 | 128 | 256 |
| Sub-steps | 2 | 2 | 2 |
| Recursive iters | 12 | 16 | 16 |
| Grad-enabled iters | 8 | 8 | 8 |
| Layer division strategies | C121 | C121 | C141 |
| Window size | 512 | 512 | 512 |
| Anchor interval | 64 | 64 | 64 |

RWKV-v7 attention parameters per layer are

$$P_{\text{RWKV}} = 4D^2 + 2D(D_{\text{DEC}} + D_{\text{GATE}}) + 6MDr + 4MDD_{\text{AAA}} + (2M + 11)D,$$

while GQA attention parameters per layer are

$$P_{\text{GQA}} = 2D(H_q d_h) + 2DK + 2d_h \quad (\text{if } H_q d_h = D, \ \ P_{\text{GQA}} = 2D^2 + 2DK + 2d_h).$$

The total trainable parameter count is therefore

$$\boxed{P_{\text{total}} = 2VD + (2D^2 + D) + 2D + L(3DD_m + 4D) + L_e P_{\text{RWKV}} + L_o P_{\text{GQA}}.}$$

With weight tying, replace $2VD$ by $VD$ and adjust if biases are added elsewhere.

## G    LLM USAGE STATEMENT

In this work, LLMs were utilized exclusively for improving the clarity of the manuscript and enhancing the code, and were not applied to any other critical components.

