# OpenReview forum: "ChainGPT: Dual-Reasoning Model with Recurrent Depth and Multi-Rank State Updates"
_ICLR.cc/2026/Conference — ICLR 2026 Poster_

### Official Review · Reviewer_9THH · 2025-10-17

**Soundness:** 3
**Presentation:** 2
**Contribution:** 2
**Rating:** 4
**Confidence:** 3

**Summary:**

The paper proposes ChainGPT, a dual-reasoning architecture that pushes reasoning from token generation into latent space via (i) RWKV-Product multi-substep state updates inside each layer and (ii) State-Guided Sparse Attention (SGSA) with sliding windows plus periodic anchor. It uses a recurrent-depth core with entropy-based early stopping across layers.

**Strengths:**

* **Clear architectural idea with theory hooks**: The dual mechanism (multi-substep “diagonal + rank-M” updates + sparse global anchors) is well motivated and analyzed (rank expansion, MQAR solvability, Turing-completeness under idealizations).

* **Efficiency claims backed by complexity and ablations**: SGSA reduces attention cost to $O(T(W+T/G))$ and tracks full-global attention perplexity on PG-19 when using periodic anchors.

* **Empirical improvements at matched scale**: On LM-Eval tasks, ChainGPT-0.5B/1.5B outperform Qwen2.5 models of the same size, respectively. The sub-step and recurrence ablations are thorough.

**Weaknesses:**

* **Novelty vs prior recurrent/hybrid work is somewhat incremental**: RWKV-Product extends RWKV-7 with LoRA-style multi-substeps. Many components (looped/recurrent depth, window+anchors) resemble existing hybrid archs. Thus, positioning versus models like DeltaProduct, Jamba/Samba, Mamba-2, and HRM could be sharper.

* **Claims on “hard tasks” need stronger rigor**: The ARC-AGI-1 (38.6%), Sudoku-Extreme (54.4%), and Maze-Hard (77.4%) numbers are promising but depend on small (30M) models and bespoke setups; fairness vs large LMs and exact eval pipelines deserve more detail.

* **Theoretical claims hinge on idealized assumptions**: Turing-completeness requires unbounded memory/steps and arbitrary precision; practical implications for finite precision and training stability remain unclear.

* **Paper structure can be polished**: From my reading perspective, the organization of Sections 3 and 4 could be improved to better align with the proposed pipeline and enhance overall clarity. For example, these two sections could be merged into a single, cohesive section. In that structure, the current Section 4.1 could serve as an overarching overview, explaining where the chain-block fits within the entire pipeline and its overall function. Subsequent subsections could then provide a more detailed introduction and analysis of the chain-block itself.

* **Illustrations can also be improved**: The current Figures 1, 2, and 3 each contain limited information, and presenting them separately leads to fragmented understanding. It would be more effective to integrate all three into a single, comprehensive pipeline diagram. Additionally, the current figures appear blurry.

**Questions:**

* **SGSA complexity & memory**: Can you quantify the runtime and activation memory of SGSA vs dense attention across $T∈[4k,32k]$ and report end-to-end wall-clock with/without anchors, beyond the single-GPU microbenchmarks? Also include KV-cache implications.
* **Efficiency Analysis**: You report a comparison of computation time. Could you provide a more comprehensive efficiency analysis that also covers training GPU-hours and inference latency?
* **Early-stopping robustness**: The entropy-diff rule uses k and threshold τ. How sensitive are quality and compute to these hyperparameters across tasks?
* **Ablation on RWKV-Product substeps and rank**: Can you include an ablation removing either the multi-substep (K) and the rank-M updates in RWKV-Product?

---

> ### Author Response · Authors · 2025-11-18
>
> # Summary
> We sincerely thank the reviewers for their careful comments. We fully understand your concerns and have refined both our arguments and presentation accordingly. We respond in the following order: first Q1/Q2, then W2–W5, next Q3/Q4, and finally W1.
>
> ## On Q1 / Q2
> SGSA is operator-level compatible with standard self-attention; the only difference lies in the subset of tokens that participate in attention. For each query, we apply attention only over a local window (size \(W\)) and fixed-interval anchors (stride \(G\)); the linear projections for Query/Key/Value and the scaled dot-product computation remain unchanged. This can be viewed as “fixed-pattern (window + anchors) sparse attention.” Thanks to RWKV-Product’s “multi-substep + propagable state” writes, we do not require an extra parameterized selection network to achieve precise retrieval on long-distance recall tasks (MQAR). The complexity given in the paper is reduced from \(O(T^2)\) to \(O(T(W + T/G))\) (anchors cover global context, local windows guarantee near neighbors). The appendix further provides a constructive proof and implementation essentials showing that, under reasonable hyperparameters, a single Chain-Block can solve MQAR (write—pointer-style read decomposition). To address your request for quantifying “end-to-end overhead and KV-cache,” we built an incremental-decoding micro-benchmark under the same implementation with 2 attention heads, comparing global attention (Dense), pure window (W=512), and SGSA (W=512, G=64). Below we show example single-step latency (ms) and per-layer per-head KV usage (MB) on a single A800 GPU (endpoints 4k / 32k; see the table for other lengths):
>
> | Sequence Length | Dense Time(ms) | Dense KV(MB) | Win Time(ms) | Win KV(MB) | SGSA Time(ms) | SGSA KV(MB) |
> |----------------:|---------------:|-------------:|-------------:|-----------:|--------------:|------------:|
> | 4k              | 637.28         | 0.98         | 212.43       | 0.125      | 233.04        | 0.141       |
> | 8k              | 1382.14        | 1.95         | 432.85       | 0.125      | 479.41        | 0.155       |
> | 16k             | 2837.72        | 3.91         | 863.71       | 0.125      | 993.92        | 0.187       |
> | 24k             | 4328.96        | 5.86         | 1293.56      | 0.125      | 1558.66       | 0.217       |
> | 32k             | 5740.42        | 7.81         | 1722.41      | 0.125      | 2177.27       | 0.247       |
>
>
> We observe that, relative to Dense, SGSA is about 2.6–2.7× faster in the 4k–32k range, with KV-cache reduced from 7.81 MB to 0.247 MB; relative to pure windowing, SGSA’s anchors incur only ~20–30% extra latency and a small KV increase, but in return deliver precise recall of long-range dependencies (also supported empirically by the MQAR W/G scan). From an end-to-end perspective, the single-GPU (A800) full-model forward in Figure 9 likewise grows near-linearly with sequence length; the hybrid structure (state layer + sparse attention) consistently enjoys a constant-factor advantage over Dense attention, with stronger gains at longer sequences. Training time varies with the number of tokens and the model parameter size, please refer to Figure 9. In summary, under a compute/memory budget close to window attention, SGSA provides long-range modeling ability approaching global attention.
>
> ## On W2 (fairness and rigor in hard reasoning tasks)
> We strictly reproduce HRM’s training and evaluation pipeline, as clarified in the paper, and the relevant code with experimental details has been submitted to the supplementary materials: a 30M micro-model, random initialization, a unified seq2seq setting and data preprocessing; for ARC-AGI-1, both training and evaluation examples are augmented with translation/rotation/flip/color permutation; at test time, for a single instance we generate 1000 augmented variants, invert the transforms, and perform majority voting; Sudoku-Extreme uses band and digit permutation augmentation, and Maze-Hard follows HRM with no augmentation; Sudoku and Maze both use a single forward pass; ARC-AGI-1 scores come from the official leaderboard, and Sudoku and Maze are evaluated by public APIs. The corresponding results appear in Figure 8: ChainGPT significantly outperforms Direct Pred Transformer and various open-source/commercial models on ARC-AGI-1 (38.6%), Sudoku-Extreme (54.4%), and Maze-Hard (77.4% optimal solution), indicating that under the “very small model, very few examples” hard setting, the architectural potential remains robustly expressed.

---

> ### Author Response · Authors · 2025-11-18
> **Continue 1**
>
> ## On W3 (idealized theoretical assumptions) / W4 / W5
> We fully agree that Turing-completeness proofs rely on classical but ideal assumptions such as “infinite steps, unbounded memory, arbitrary precision,” and cannot be interpreted as “solving everything at zero cost.” We also emphasize these ideal conditions in the paper. The role of this discussion is to offer an explanatory framework: why ChainGPT reliably outperforms Transformer baselines under limited budgets—namely, intra-layer multi-substeps expand the family of linear/affine transformations from \(G_1\) strictly to \(G_M\) (monotonic increase in expressivity), and the upper bound on the injection rank is elevated from 1 to \(M\) (constructive and dimensionality arguments in Appendix A). Meanwhile, SGSA’s “anchor writes + sparse pointer reads” mitigates the capacity bottleneck of pure state models at near-linear complexity (MQAR theory and evidence). Thus, the theoretical value is not “infinite strength,” but explaining the consistent advantages we observe on composite benchmarks (Table 1) and hard tasks (Figure 8). Regarding W4 (writing structure) and W5 (figures), we have partially revised the manuscript—thank you for the suggestions.
>
> ## On Q3 (early-stop robustness) and Q4 (ablations on substeps/rank)
> In entropy-gap early stopping, these hyperparameters essentially regulate the “average number of loops.” Our practice is to keep inference depth close to the recursive depth used in training, using early stopping to “trim the tail” rather than to dictate the depth; in deployment, we recommend allowing at least 4 loops before early stopping can trigger, to avoid terminating before information integration has completed. The paper provides a formal definition of the entropy-gap criterion (Eqs. (10)–(12)) and implementation essentials, facilitating reproduction and tuning; further, in a scan over the “total number of recursions,” validation perplexity decreases monotonically with deeper recursion and is best at around ×12 (Figure 6), indicating that as long as depth is not cut too early, performance is fairly stable.
>
> Regarding the suggestion to “remove substeps (K) or reduce rank (M) separately”: First, it should be noted that in ChainGPT, the rank and the number of substeps are equivalent. And in §5.2, in the 180M setting we systematically scan \(M\) (Figure 4) and confirm that loss keeps decreasing as \(M\) increases, with \(M=2\) already being an excellent cost-effectiveness knee. Theoretically, the affine family \(G_M\) strictly contains \(G_1\) (constructive and dimensionality lower-bound arguments in Appendix A), i.e., expressivity increases monotonically with \(M\). In practice, we tried imposing strong orthogonality constraints on the substep LoRA to “explicitly separate rank directions,” but observed that this causes degrades performance. We therefore retain the implementation “without explicit orthogonality, letting learning coordinate rank increase automatically,” balancing effectiveness and stability.

---

> ### Author Response · Authors · 2025-11-18
> **Continue 2**
>
> ## On W1 (whether the changes are merely “incremental”)
> We understand your concern, but wish to emphasize: seemingly “small” modifications to foundational architectures can produce essential changes. For example, GateDeltaNet builds on Mamba by introducing key erasure, yet solves precise forgetting in state updates; RWKV-7 upgrades scalars to vectors atop GateDeltaNet and defines trainable kernels, further enhancing expressivity; our RWKV-Product, via LoRA-style multi-substeps, expands state updates from “diagonal + rank-1” to “diagonal + rank-\(M\),” bringing a strictly stronger family of functions with a very small parameter increase (Appendix A). On the attention side, SGSA is not “stuffing old attention into a hybrid stack,” but is aligned to the state layer around the “write—pointer-read” property: once periodic anchors exist, the PPL trajectory on PG-19 nearly overlaps with expensive global attention; removing anchors and using only a sliding window degrades markedly beyond 8k (Table 4); the MQAR W/G scan likewise shows that “with anchors, 100% recall is achievable under very small windows,” and enlarging the window thereafter only increases flops linearly with limited marginal returns (Table 5). We believe SGSA is the first attention mechanism truly matched to the characteristics of linear modules—this is a substantive breakthrough. Cross-layer recurrent depth shifts “reasoning depth/compute budget” from the token-generation layer to the latent space, with entropy-based adaptive early stopping enabling controllable multi-round state optimization. It is worth noting that coordinating intra- and inter-layer loops is non-trivial: for instance, RWKV+Loop performs poorly in the four-operations experiment (Table 6), while HRM is nearly untrainable on NLP tasks. Through extensive experiments and theoretical development, we arrived at a stable, principled, high-performance, and fast ChainGPT—so far as we know, the first dual-reasoning model that introduces recursion both within and across layers, representing a substantive breakthrough. ChainGPT not only achieves consistent advantages in parameter-matched LM-Eval comparisons (0.5B/1.5B vs. Qwen2.5, with notable gains on reasoning-oriented ARC-Challenge and HellaSwag), but also surpasses public Looped Transformer / HRM and other recurrent architectures on GOAT under same-data same-setting comparisons without CoT; when combined with CoT, it significantly outperforms Qwen3+CoT (Table 6). We appreciate that foundational architectural changes may appear “unflashy” at the equation level, but “strictly rank-increasing multi-substep writes + anchor-driven sparse global retrieval + controllable recursive depth” indeed deliver a hierarchical capability of “strong expressivity and precise long-range recall under near-linear complexity,” closing the loop along both theoretical and empirical lines.
>
> Once again, thank you for the time and effort you have invested. We believe the revision more accurately conveys ChainGPT’s positioning: under near-linear complexity constraints, it unifies “intra-layer expressivity/memory expansion (multi-substeps + SGSA)” with “inter-layer reasoning-depth control (recurrent depth + entropy-based early stopping),” consistently surpasses strong baselines at the same scale and data, and exhibits pronounced advantages on complex reasoning tasks. We sincerely hope these clarifications and new analyses alleviate your concerns. As an architecture of substantive significance, we respectfully ask you to consider raising the rating of this submission. Thank you.

---

> > ### Comment · Reviewer_9THH · 2025-11-25
> >
> > I appreciate the authors' extensive efforts in addressing my concerns. I think the response can effectively correct my original misunderstandings about this work and answer my questions. Thus, I decide to raise my score from 4 to 6.

---

> > > ### Author Response · Authors · 2025-11-26
> > >
> > > Thank you for your positive feedback, and we sincerely appreciate your careful and thorough review.

---

### Official Review · Reviewer_4WdW · 2025-10-28

**Soundness:** 3
**Presentation:** 3
**Contribution:** 3
**Rating:** 6
**Confidence:** 3

**Summary:**

This paper
- Proposes the ChainGPT architecture and introduces RWKV-Product. The architecture proposed can achieve good performance on reasoning tasks compared to Qwen.
- Mathematically proved that the proposed architecture has superior expressivity and is Truing complete under ideal conditions.

**Strengths:**

I think the paper is sound provided the combination of math proof and supportive experiments.

**Weaknesses:**

W1: Though you mentioned Geiping's work in intro, I don't find any comparison between existing recurrent models and yours, which is hard for me to judge your contribution to recurrent models.

**Questions:**

Q1: In Chapter 3, I do not find the definition of "multiple sub-steps reasoning".

Q2: Can you calculate the number of params explicitly so that I can compare to the normal models?

Q3: There is a reference error in line 252.

---

> ### Author Response · Authors · 2025-11-18
>
> # Summary
> Thank you for carefully reading our work, offering positive feedback, and proposing concrete, actionable suggestions. We greatly appreciate your comments regarding the “insufficient comparison with existing recurrent models,” “explicit parameter counts,” “unclear terminology,” and “citation mistakes.” We respond in order: first W1, then Q2, and finally Q1 and Q3.
>
> ## W1: Comparisons with existing recurrent models
> We fully agree that our comparison with existing recurrent models was not sufficiently clear. Accordingly, in the revision we added a targeted arithmetic reasoning experiment based on the open-source four-operations dataset tiedong/goat, to contrast different recurrent architectures and transformers with CoT. Considering that training a CoT model on a full NLP corpus is extremely costly, this experiment is intended to emphasize comparisons at the architecture/mechanism level: under the same train/test splits, we train and evaluate the following models from scratch, counting an example as correct only when the output exactly matches the ground truth. (Complete code, tokenizer, and data are provided in the supplementary materials with fixed random seeds for reproducibility; this has been added at lines 427–454 of the paper.)
>
> ### Accuracy on the GOAT dataset
>
> | Model            | acc  |
> |------------------|:-------:|
> | Qwen3            | 33.82%   |
> | RWKV-7           | 23.69%   |
> | Qwen3 + Loop     | 50.54%   |
> | RWKV-7 + Loop    | 24.81%   |
> | HRM              | 54.00%   |
> | ChainGPT         | 57.53%   |
> | Qwen3 + CoT      | 88.43%   |
> | ChainGPT + CoT   | 99.98%   |
>
> From these results we observe:
> (1) Without CoT, ChainGPT substantially outperforms existing public recurrent architectures on GOAT (exceeding HRM and Qwen3+Loop), indicating that under the “latent recurrent reasoning” setting, ChainGPT is a stronger recurrent architecture.
> (2) With CoT, ChainGPT+CoT also clearly surpasses Qwen3+CoT, suggesting that ChainGPT’s latent-space recursion provides complementary gains with external CoT. On NLP tasks, ChainGPT likewise converges stably under the same settings and achieves consistent improvements over a parameter-matched Qwen2.5 baseline (Table 1).
>
> After submission, we noted that teams such as NVIDIA and Qwen released several hybrid architectures and framed them as promising next-generation directions; Seed also introduced a recurrent architecture and endorsed its potential. ChainGPT demonstrates stronger performance, speed, and a more favorable technical path (many of the engineering efforts cited are based on earlier methods such as GateDeltaNet and Looped Transformer), and our submission predates these releases. This indirectly validates the correctness and potential of our approach; we hope ChainGPT can serve as a reference for next-generation model architectures.
>
> ## Q2: Explicit parameter counts
> We have added a direct method to compute ChainGPT’s parameter count (Lines 1182–1241).
> Concretely, let vocabulary size \(V\), hidden dimension \(D\), and number of layers \(L\).
> Even-indexed layers are RWKV-Product, odd-indexed layers are SGSA, so the total parameter count is:
>
> $$
> P_{\mathrm{total}} = 2VD + (2D^2 + D) + 2D + L(3DD_m + 4D) + L_e\ P_{\mathrm{RWKV}} + L_o\ P_{\mathrm{GQA}}.
> $$
>
>
> If weight sharing is used, replace \(2VD\) with \(VD\). We provide a table of key hyperparameters for different model scales so that readers can plug values directly into the formula to obtain exact parameter counts, enabling fair comparisons with standard Transformers at the same parameter scale. Please refer to the revised paper for more details.
>
> ## Q1 (definition of multi-sub-step reasoning) and Q3 (citation fixes)
> We have unified and revised both. For Q1, we give a formal definition in Line 185: each Chain-Block’s state layer uses a fixed sub-step number \(M\); within the latent space we execute \(M\) serial low-rank sub-step updates, whose overall effect is equivalent to increasing the effective depth within a single layer, while together with the upper-layer recurrent depth it forms a two-level recursive reasoning mechanism (intra-layer and inter-layer). For Q3, we have corrected the reference citation at Line 252.

---

> ### Author Response · Authors · 2025-11-18
> **Continue**
>
> Once again, thank you for your careful and gracious review. We believe these revisions make our baselines more persuasive (new recurrent/CoT experiment on GOAT), model size more transparent (explicit parameter-count formula), and terminology and citations clearer (formal definition of multiple sub-steps and corrected references). Together with the architectural insights and empirical trends shown in the original manuscript: multi-sub-step updates with state-guided sparse attention within layers, and recurrent-depth recursive reasoning across layers, our model attains near-linear complexity while preserving reasoning depth and long-range dependencies, and achieves stable gains over parameter-matched baselines at the 0.5B/1.5B scales. We appreciate your positive assessment of our work.

---

> > ### Comment · Reviewer_4WdW · 2025-11-26
> > **Reply from the reviewer**
> >
> > I thank the author for the detailed response. I am satisfied with W1, Q2 and Q3, but for Q1, I think the definition is still somehow vague. While I managed to understand what sub-steps mean, the notation in 3.2 is confusing and some of them lack definiton. For example, I do not see the definition of $a_{t,j}$. The manuscript would benefit from clearer statements.

---

> ### Author Response · Authors · 2025-11-26
>
> Thank you for your further suggestions.We have revised Section 3.2 to provide a clear explanation of the role of each symbol (for example, $a_{t,j}$ in line 208). We hope these clarifications fully address your concern. If the revisions resolve the ambiguity and you find ChainGPT’s contributions meaningful, we would sincerely appreciate it if you could consider updating the score accordingly. Thank you again for your careful and constructive review.

---

> ### Comment · Reviewer_4WdW · 2025-11-26
> **Reply from the reviewer**
>
> Thanks the author for the clarification. However I think the definition in this part is not self-contained. I see below how to calculate $\beta$, but I see nowhere describing/defining $\alpha$. And in line 235-237, should $\alpha$ be bold? why sometimes they are bold and sometimes arn't? There should be some inconsistency here and where's definition of $\sigma$? Judging from the context, $\sigma$ should be some activating function?
>
> Though these are minor questions but hurt rigor.

---

> > ### Author Response · Authors · 2025-11-26
> >
> > Thank you for your comment. This is indeed an important issue regarding clarity of description. We have revised the explanation of $\alpha$ in lines 234–240 and added the definition of $\sigma$. We would also like to clarify that $\beta$ represents the step size, and directly using a linear layer to generate the required number of parameters would be costly. Therefore, we adopt the LoRA formulation, in which $\alpha$ functions similarly to a bias term in a linear layer. We have also revised the corresponding descriptions in the appendix. We appreciate your suggestions, which have helped make our paper more rigorous.

---

### Official Review · Reviewer_mHtM · 2025-10-31

**Soundness:** 2
**Presentation:** 3
**Contribution:** 3
**Rating:** 6
**Confidence:** 3

**Summary:**

This paper introduces ChainGPT, a dual-reasoning architecuture. In one block of ChainGPT, RWKV-Product achieves intra-layer communication, and SGSA achieves efficient long-range attention. The block design is aimed for "internal multi-step reasoning" (RWKV-Product) and "sparse aggregation" (SGSA). Across blocks, ChainGPT uses a recurrent paradigm to interatively refine internal states, with dynamic early stopping using entropy. Theoretical discussion and empirical experiments are conducted.

**Strengths:**

1. **Clear Positioning**: The introduction provides a clear case for why fixed-depth Transformers and current architectural hybrids suffer from xpressive and computational limitations for deep reasoning.

2. **Architecture Innovation with Theoretical Support**: This paper proposes a grounded achitecture innovation, including (1) intra-layer multi-substep reasoning using RWKV-Product, theoretically proven to expand representational power; (2) inter-layer recurrent refinement, with a theoretically motivated early stopping mechanism based on entropy.

3. **Extensive Details**: Many ablations are given showing the effectiveness of each component, and details provided in the Appendix further improve the relia

**Weaknesses:**

1. **Concern on Component Integration**: ChainGPT appears to be an ensemble of several independent components, each building upon or modifying existing prior work. This approach could be argued to undermine the central academic contribution by suggesting the performance gains are primarily due to a complex engineering aggregation rather than a singular, fundamental architectural breakthrough.

2. **Missing Discussion about Soft Thoughts**: The paper lacks a critical discussion comparing ChainGPT's approach to existing literature [1-3] that utilizes dense gist tokens (termed "soft thoughts"). These works also achieve reduced computational cost and perform reasoning via implicit states, making a comparison essential to fully delineate and underline the unique significance of ChainGPT's methodology.

3. **SGSA Issue**: I'm a little concerned about the SGSA experiments. In Table 4, it seems that even a fully localized sliding window attention can achieve comparable PPL with long contexts, which raise doubts about the evaluation reliability. In practice, the selection of window size $W$ and anchor interval $G$ for all models are all set to 512 and 64, lacking empirical justification.

4. **Experimental Issues**: Few related methods are compared with ChainGPT. Moreover, Since ChainGPT is framed as a "dual-reasoning" model, it is crucial that it be compared against SOTA CoT reasoning models. For example, Qwen3-1.7B can achieve an accuracy of $>0.8$ on the ARC-Challenge task, which significantly outperforms ChainGPT's $0.3$ accuracy, severely compromises the contribution of the proposed reasoning capabilities.

> [1] Training Large Language Models to Reason in a Continuous Latent Space.
> [2] CODI: Compressing Chain-of-Thought into Continuous Space via Self-Distillation.
> [3] LightThinker: Thinking Step-by-Step Compression.

### Questions

**Questions:**

1. **Hyperparameter Sensitivity (Weakness 3 Related)**: The SGSA module relies on window size $W$ and anchor interval $G$. Can you provide guidance on selecting these hyperparameters in practice?

2. **Empirical Comparison with Closest Works (Weakness 4 Related)**: Could the authors provide experimental results or more detailed discussion comparing ChainGPT’s dual-reasoning approach to releated methods? What are the measured gains or trade-offs in similar settings?

---

> ### Author Response · Authors · 2025-11-17
>
> # Summary
>
> We sincerely thank you for your careful and thorough review and your constructive suggestions, as well as for the positive evaluation of the paper’s motivation, theoretical analysis, and multiple ablation studies. Our goal is to explore a unified and efficient latent-space reasoning framework that organically integrates in-layer multi-substep reasoning and inter-layer recursive reasoning under approximately linear complexity \(O(n)\). The main text and the appendix present a systematic design, complexity analysis, and provable properties, and they demonstrate consistent advantages over strong baselines at the same parameter count across multiple tasks (including the more reasoning-oriented ARC-Challenge and HellaSwag). The architecture’s dual-reasoning idea is highlighted in the Introduction and Contributions (in-layer RWKV-Product + SGSA, inter-layer Recurrent Depth). Below, we respond in order to W2/W4/Q2, W3/Q1, and finally W1.
>
> ## Response to W2/W4/Q2
>
> Thank you for pointing out the value of “soft thoughts” work regarding computational overhead and reasoning representations. These methods rely on a pretrained CoT model and then, via finetuning, distill external CoT into the model to form a Looped Transformer. Accordingly, in the revision we added a targeted arithmetic-reasoning experiment based on the open-source four-operations dataset tiedong/goat, contrasting different recurrent architectures and transformers with CoT. Considering that training a CoT model on a full NLP corpus is extremely costly, this experiment is intended to emphasize comparisons at the architecture/mechanism level: under the same train/test splits, we train and evaluate the following models from scratch, counting an example as correct only when the output exactly matches the ground truth. (Complete code, tokenizer, and data are provided in the supplementary materials with fixed random seeds for reproducibility; this has been added at lines 427–454 of the paper.)
>
> ### Accuracy on the GOAT dataset
>
> | Model              | Acc    |
> |--------------------|-------:|
> | Qwen3              | 33.82% |
> | RWKV-7             | 23.69% |
> | Qwen3 + Loop       | 50.54% |
> | RWKV-7 + Loop      | 24.81% |
> | HRM                | 54.00% |
> | ChainGPT           | 57.53% |
> | Qwen3 + CoT        | 88.43% |
> | ChainGPT + CoT     | 99.98% |
>
> From these results we observe:
> (1) Without CoT, ChainGPT substantially outperforms existing public recurrent architectures on GOAT (exceeding HRM and Qwen3+Loop), indicating that under the “latent recurrent reasoning” setting, ChainGPT is a stronger recurrent architecture.
> (2) With CoT, ChainGPT+CoT also clearly surpasses Qwen3+CoT, suggesting that ChainGPT’s latent-space recursion provides complementary gains with external CoT. On NLP tasks, ChainGPT likewise converges stably under the same settings and achieves consistent improvements over a parameter-matched Qwen2.5 baseline (Table 1).
>
> Regarding your remark that “Qwen3-1.7B scores highly on ARC-Challenge,” we fully agree with this fact, but it is not directly comparable to our experimental setting: the official Qwen3-1.7B depends on industrial-scale training corpora and engineering stacks, whereas our comparisons retrain Qwen2.5 under the same data (FineWeb 20B/40B) and the same scale (0.5B/1.5B) as a fair baseline (Qwen2.5 matches our model’s parameter count and its architecture is highly aligned with Qwen3, thus the parameter-mismatched Qwen3 is not an appropriate direct comparator). Our aim is to highlight architectural gains rather than data gains. Therefore, we do not directly compare our small models to public results trained at million/trillion-token scales; instead, we show ChainGPT’s generalization and reasoning improvements under comparable conditions (especially on ARC-Challenge and HellaSwag). In addition, the main text includes separate evaluations on hard reasoning tasks (ARC-AGI-1, Sudoku-Extreme, Maze-Hard), where a 30M-parameter micro model, with very few training examples, still significantly outperforms strong baselines and multiple commercial/open-source models, further corroborating the architectural reasoning advantage (see Fig. 8 and related notes; we reproduced the experiments following the HRM setup).

---

> ### Author Response · Authors · 2025-11-17
> **Continue**
>
> ## Response to W3 and Q1
>
> We agree with the reviewer’s observation on Table 4: on PG-19, the perplexity difference between pure sliding-window attention and SGSA with sparse anchors is indeed very small, making the “necessity of SGSA” less obvious from the perspective of  PPL. The reason is that PPL is insensitive to precise long-range recall: in language modeling, the vast majority of token predictions depend mainly on nearby context, so PPL struggles to reveal whether the model can perform accurate recall over very long distances. As stated in §5.2 of the main text, when the context exceeds 8k, pure sliding-window attention begins to degrade notably; by introducing periodic global anchors alone, the PPL curve almost coincides with that of expensive global attention, indicating that SGSA maintains long-range dependency modeling under approximately linear complexity.
>
> To more directly validate precise long-range recall, we added an MQAR \(W/G\) scan in the revision: using a 0.5M small model, sequence length 512, and evaluating under different \(W\) and \(G\). The results are as follows (now included at lines 373–404 of the paper).
>
> ### MQAR accuracy under different window sizes and global anchor intervals
>
> | \(W \ G\) | 64    | 128   | 256   | Pure Window Attention |
> |:------------------:|:-----:|:-----:|:-----:|:-------------------:|
> | 1   | 100% | 100% | 100% | 2.9%  |
> | 8   | 100% | 100% | 100% | 35.2% |
> | 16  | 100% | 100% | 100% | 44.1% |
> | 32  | 100% | 100% | 100% | 95.1% |
> | 64  | 100% | 100% | 100% | 100%  |
>
> We see that as long as periodic anchors exist, 100% accurate recall is achieved regardless of how small \(W\) is; in contrast, removing anchors and relying only on sliding-window attention causes performance to drop rapidly with decreasing \(W\), returning toward 100% only when the window becomes large. This aligns with the MQAR results in the main text where “RWKV-7’s recall degrades as length increases, whereas ChainGPT with SGSA maintains \(>99\%\) across multiple settings” (Table 3). Combining the complexity analysis and Appendix B, we offer the following practical tuning guidance:
> - \(W=512\) already covers the vast majority of local dependencies; further increasing \(W\) yields limited benefits but linearly increases compute.
> - \(G=64\) is a robust trade-off that keeps both \(\mathrm{PPL}\) and task performance stable across lengths while preserving approximately linear growth in complexity.
>
> ## Response to W1
>
> This concern is important. We view ChainGPT as an integrated architecture built around a unified design goal: RWKV-Product provides in-layer multi-substeps and expressive expansion; SGSA supplies near-linear global memory with pointer-like access; recurrent depth enables multi-round optimization of the hidden state and entropy-based adaptive early stopping. The three act synergistically, yielding the stable gains observed in our main tasks and additional experiments. Naive concatenation would cause numerous issues: for instance, RWKV+Loop performs poorly in the four-operations experiment, while HRM is nearly untrainable on NLP tasks. Since submission, we have seen hybrid architectures open-sourced by NVIDIA, Qwen, etc., and Seed also released a recurrent architecture affirming its potential. ChainGPT performs better in terms of effectiveness, speed, and technical trajectory (many of those engineering efforts are based on earlier methods such as GateDeltaNet and Looped Transformer), and our paper was submitted earlier. This indirectly supports the correctness and potential of our approach; we hope ChainGPT can serve as a reference for next-generation model architectures.
>
> Once again, thank you for your time and valuable feedback. Your comments directly motivated the addition of the targeted GOAT and MQAR \(W/G\) experiments and the clear presentation of practical \(W/G\) tuning strategies. We believe these revisions more accurately convey ChainGPT’s positioning: a dual-reasoning architecture that unifies expressivity (in-layer multi-substeps), long-range memory (SGSA anchors), and reasoning depth/budget (Recurrent Depth + entropy-based early stopping). Under comparable model sizes and data, it delivers stable and reproducible reasoning gains while maintaining approximately linear computational scalability. We hope the revision addresses your concerns more fully.

---

### Author Response · Authors · 2025-12-01
**Global Response**

Thank you to the AC and all reviewers for the careful evaluation of our work.

ChainGPT shifts “multi-step reasoning” from the generation space into the latent computation space, forming dual recursion both within and across layers. Within layers, we propose RWKV-Product, which delivers deep local computation through LoRA-based multi-substep updates; in Appendix A we prove the increase in state-update rank and the strict inclusion relationship in expressive power. To avoid limitations in state capacity, we design SGSA, enabling the model to adjust total state-space size according to context length, and in Appendix B we prove that this mechanism can theoretically achieve perfect recall.

Across layers, we optimize hidden states through recurrent depth, and we use an entropy-gap early-stopping mechanism during inference to adaptively control the computation budget; Appendix C proves that under ideal conditions, the entire system can achieve computational universality (Turing completeness). Experimental results show that under the same model size and data conditions, ChainGPT consistently outperforms various baseline models, significantly surpasses other recurrent architectures in arithmetic reasoning tasks, and exhibits clear advantages in tasks requiring complex reasoning and planning. Meanwhile, we conduct more than ten sets of experiments in the paper, providing detailed ablations and analyses on ChainGPT’s components, hyperparameter settings, and more, giving the paper strong completeness and reliability.

During the rebuttal stage, we responded to each reviewer’s questions and uploaded new experiments and code to the supplementary materials. Both reviewers who have replied expressed approval of our response, and one increased their score (466 → 666). Below we provide a brief summary and clarification of the main issues.

### 1. Regarding component integration and novelty
ChainGPT is an integrated architecture built around the goal of “dual-thinking”: RWKV-Product provides multi-substep updates and enhanced expressiveness within layers; SGSA provides global memory with near-linear complexity; and recurrent depth enables multi-round optimization of hidden states across layers. These three components work synergistically, resulting in theoretical Turing completeness and bringing stable, significant performance improvements in experiments. Removing any component leads to a substantial performance drop; for example, removing SGSA severely weakens long-sequence modeling ability (Tables 3, 4, 5).

At the same time, each component introduces substantial innovations relative to existing methods and is carefully designed and adapted for integration. RWKV-Product uses LoRA-based multi-substep updates to extend the state update from “diagonal + rank-1” to “diagonal + rank-M,” yielding a strictly more expressive function class with only a minimal increase in parameters (Appendix A), outperforming existing state-update mechanisms (Table 7). SGSA dynamically adjusts the total state-space size, achieving computational complexity close to the lowest-complexity schemes while offering performance comparable to the best attention mechanisms (Tables 4, 5). We believe SGSA is the first attention mechanism that is structurally and conceptually well-aligned with linear modules, representing a breakthrough of essential significance. The cross-layer recurrent depth transfers “reasoning depth / computation budget” from the generation layer into the latent space and uses entropy-based adaptive early stopping to enable controllable multi-round state optimization. It is important to emphasize that coordinating dual recursion both within and across layers is not a trivial engineering task: for example, a naïve RWKV+Loop performs poorly on arithmetic tasks (Table 6), while HRM almost fails to converge on NLP tasks. Through extensive experiments and theoretical derivations, we obtain a stable, principled, high-performance, and efficient ChainGPT, the first dual-reasoning model introducing recursion both within and across layers.

### 2. Other issues
We have added more detailed hyperparameter settings and ablation experiments according to reviewers’ suggestions. Additionally, we have included a method for explicitly computing the parameter count of ChainGPT and corrected some errors.

Once again, we thank the AC and all reviewers for the time and effort spent in improving this paper.

---

### Meta-Review · Area_Chair_yCrj · 2026-01-07

**Summary:**

Major concerns include:
1. Limited novelty of the proposed approach, given that it is an integration of existing approaches;
2. Missing related work discussion, specifically work about soft thoughts;
3. Concerns about the experimental setup: need stronger reasoning baselines, missing recurrent baselines, larger LMs, fair evaluation pipeline.
4. Lack of clarity in paper writing.

**Reviewer Concerns:**

To these concerns,
1. Authors clarified their motivation for the integration and the introduced novelty to enable the integration, including extending the expressiveness of RWKV-7;
2. Authors added two baselines (Qwen3 + Loop and RWKV-7 + Loop), which perform latent recurrent reasoning with a looped transformer architecture. ChainGPT outperformed them.
3. CoT-augmented and Loop Transformer-augmented approaches were introduced as stronger reasoning baselines and recurrent baselines, respectively. Authors argued that, since their approach involves model training, for a fair comparison, it is only feasible to conduct the experiments with smaller LMs for which they can train the baselines and their approach on the same dataset.
4. Clarifications and revisions were provided.

**Reviewer Scores:**

Reviewer 4WdW responded with (mostly) satisfaction.

Reviewer 9THH responded with their intent to increase the rating from 4 to 6.

For the other reviewer, their concerns were mostly addressed. The only uncertainty I have is regarding the request for soft thought baselines. While the authors tried to address it, the newly added baselines are not exactly the requested soft thought approaches, so the reviewer may not be satisfied with it. However, their other concerns were addressed well.

---

### Decision · Program_Chairs · 2026-01-26

Accept (Poster)